



**Representation of a two-way coupled irrigation system in the Common Land Model**
Shulei Zhang[1, *, #], Hongbin Liang[1, *, #], Fang Li[2], Xingjie Lu[1], Yongjiu Dai[1]
*[1] Southern Marine Science and Engineering Guangdong Laboratory (Zhuhai), School of*
*Atmospheric Sciences, Sun Yat-sen University, Guangzhou, China*
*[2]International Center for Climate and Environment Sciences, Institute of Atmospheric Physics,*
*Chinese Academy of Sciences, Beijing, 100029, China*
*[*] **Corresponding Authors:** Shulei Zhang (zhangshlei@mail.sysu.edu.cn) & Hongbin Liang*
*(lianghb25@mail2.sysu.edu.cn)*
*[#] These authors contributed equally to this work.*
**Abstract**
Human land–water management, especially irrigation water withdrawal and use, significantly
impacts the global and regional water cycle, energy budget, and near-surface climate. While land
surface models are widely used to explore and predict the impacts of irrigation, the irrigation
system representation in these models is still in its early stages. This study enhances the
Common Land Model (CoLM) by introducing a two-way coupled irrigation module. This
module includes an irrigation water demand scheme based on soil moisture deficit, an irrigation
application scheme considering four major irrigation methods, and an irrigation water
withdrawal scheme that incorporates multiple water source constraints by integrating CoLM with
a river routing model and a reservoir operation scheme. Crucially, it explicitly accounts for the
feedback between irrigation water demand and supply, which is constrained by available surface
water (i.e., runoff, streamflow, reservoir storage) and groundwater. Simulations conducted from
2001 to 2016 at a 0.25° spatial resolution across the contiguous United States reveal that the
model effectively reproduces irrigation withdrawals, their spatial distribution, and water source
proportions, aligning well with reported state-level statistics. Comprehensive validation
demonstrates that the new module significantly improves model accuracy in simulating regional
energy dynamics (sensible heat, latent heat, and surface temperature), hydrology (river flow),
and agricultural outputs (yields for maize, soybean, and wheat). Application analyses highlight
the potential of the enhanced CoLM as a valuable tool for predicting irrigation-driven climate
impacts and assessing water use and scarcity. This research offers a pathway for a more holistic
representation of fluxes in irrigated areas and human-water interactions within land surface
models. It is valuable for exploring the interconnected evolution of climate, water resources,
agricultural production, and irrigation activities, while supporting sustainable water management
decisions in a changing climate.





## 1. Introduction

Freshwater resources are indispensable for human society. Since 1900, the global population has increased more than fourfold, leading to a nearly sixfold rise in water withdrawals, from approximately 500 km³ per year in 1900 to about 3000 km³ per year in 2000, with agriculture being the dominant water user (Pokhrel et al., 2016). Around 70% of global freshwater has been withdrawn for irrigation (Campbell et al., 2017), accounting for 90% of consumptive water use (Siebert and Döll, 2010), with irrigated areas providing approximately 40% of global food production on just 2.5% of global land (Abdullah, 2006). Accompanied by significant socioeconomic benefits, these intense human land-water management practices have profoundly altered Earth's surface and impacted terrestrial water and energy cycles (Ketchum et al., 2023; Nocco et al., 2019; Rappin et al., 2022; Thiery et al., 2017; de Vrese et al., 2016). The demand for irrigation water is anticipated to rise with the growing global population and food demand, while climate-warming-induced droughts are likely to exacerbate this need (McDermid et al., 2023; Mehta et al., 2024; Yang et al., 2023). Therefore, understanding and quantifying the impacts of irrigation water management in human-Earth system interactions are crucial for developing strategies to sustainably manage these resources amidst changing climatic and demographic conditions.

Irrigation practices transfer water from various sources, such as rivers, lakes, reservoirs, and aquifers, into agricultural systems, directly affecting the magnitude and timing of runoff and river flow (Ketchum et al., 2023). The rising irrigation demand has spurred increased construction of reservoirs and diversions, resulting in both local and downstream impacts. In some regions, water extraction for irrigation has reduced the availability of both surface and groundwater (Döll et al., 2014). Besides modifying water fluxes, irrigation also influences regional climates both locally and remotely. Locally, it alters surface albedo, evapotranspiration, and surface soil moisture, impacting regional radiation and energy balances and affecting temperature, humidity, and precipitation through land-atmosphere feedback (Chen and Dirmeyer, 2019; Kang and Eltahir, 2018; Li et al., 2022; McDermid et al., 2017; Nocco et al., 2019). Remotely, it affects climate through complex interactions between altered temperature and moisture gradients and larger-scale processes such as atmospheric circulation and wave activity (Douglas et al., 2009; Phillips et al., 2022; de Vrese et al., 2016).

Earth system models (ESMs) are powerful tools for examining the interactions and feedback among the intricately intertwined processes of the Earth system, both in the past and future. Land surface models (LSMs) are a crucial component of ESMs. Due to the complex dynamics of natural hydrological processes and anthropogenic activities, describing human-water interactions has been recognized as a significant challenge in Land surface modeling (Nazemi and Wheater, 2015). In recent years, targeted efforts have aimed to address this deficiency, yet water use remains largely underrepresented or in a nascent stage within LSMs (Blyth et al., 2021; Taranu et al., 2024). Meanwhile, global hydrological models (GHMs), originally designed for water



resource assessment, have undergone continuous improvements over the last three decades to
explicitly represent human water use (Hanasaki et al., 2018; Liang et al., 1994; Müller Schmied
et al., 2021; Sood and Smakhtin, 2015; Sutanudjaja et al., 2018; Tang et al., 2007). These models
enable the determination of the spatial distribution and temporal evolution of water resources and
water stress for both humans and other biota under the pressures of global change (Döll et al.,
2018; Schewe et al., 2014; Schlosser et al., 2014). These advancements have offered valuable
insights for incorporating human water use into LSMs.
Parameterizing irrigation water use and modeling its impacts in GHMs and LSMs has been
approached using different assumptions and simplifications in three key aspects: irrigation
demands, irrigation methods, and irrigation water supplies/withdrawals. The first aspect is
estimating irrigation water demands. Models estimate these demands using either a root-zone
soil moisture deficit approach or a crop-specific potential evapotranspiration approach. The root-
zone soil moisture deficit approach estimates irrigation demand as the water needed to keep root-
zone soil moisture (usually within the top meter of soil) above a certain threshold during the
growing season (normally a certain percentage of field capacity or soil saturation) (Ozdogan et
al., 2010). The crop-specific potential evapotranspiration approach estimates irrigation needs
based on the difference between crop-specific potential evapotranspiration and simulated
unirrigated evapotranspiration, or between potential and effective precipitation under well-
watered conditions where crops transpire at their maximum rate (Müller Schmied et al., 2021).
Notably, LSMs generally do not use potential evapotranspiration to estimate irrigation demand.
The second aspect concerns the representation of irrigation methods. Many models simplify
irrigation application by directly modifying soil moisture or treating it as additional rainfall
across all irrigated land, overlooking the diversity of irrigation techniques employed in various
parts of the world or by different farmers (Li et al., 2024; Lu et al., 2015; de Vrese et al., 2018).
Recently, some models have started integrating specific irrigation techniques for certain crops or
regions. For instance, LPJmL includes sprinkler, drip, and surface irrigation methods, and CLM
incorporates drip, sprinkler, flood, and paddy irrigation methods (Jägermeyr et al., 2015; Yao et
al., 2022). Different irrigation techniques affect farmland hydrological processes and irrigation
efficiency in distinct ways. For example, drip and surface irrigation methods avoid interception
losses observed with sprinkler (Nair et al., 2013).
Third is the representation of irrigation water supplies/withdrawals, which is particularly critical
as it involves the interaction between multiple processes or modules, such as hydrological and
agricultural systems. However, explicit representation of these interactions remains largely
absent in LSMs, despite the extensive modeling experience provided by GHMs. Such modeling
first requires identifying the sources of irrigation water, typically categorized into surface water
and groundwater. Surface water sources are normally constrained by available runoff,
streamflow, and storage such as lakes and reservoirs. Accessing these sources, such as rivers and
reservoirs, necessitates coupling with river routing and reservoir modules, which are well-



represented in many GHMs (Biemans et al., 2011; Hanasaki et al., 2018). Groundwater is
typically divided into renewable sources (baseflow or dynamic groundwater levels) and
nonrenewable sources (fossil groundwater). Some models assume an inexhaustible supply of
nonrenewable groundwater to meet irrigation demands, neglecting irrigation shortages caused by
water scarcity (Zhou et al., 2020). Additionally, some GHMs incorporate alternative sources,
such as inter-regional water transfers and seawater desalination (Hanasaki et al., 2018;
Sutanudjaja et al., 2018). A second critical aspect of irrigation water supply modeling is
determining the allocation of irrigation water among different sources, including the
prioritization of water usage. Various models adopt different assumptions for this allocation. For
example, H08 prioritizes surface water (Hanasaki et al., 2018), while WBMplus prioritizes
reservoirs and groundwater (Wisser et al., 2010). PCR-GLOBWB uses an empirical approach
that allocates groundwater use based on comparisons between baseflow conditions and long-term
historical climatology, capturing feedback between water supply and demand (Sutanudjaja et al.,
2018). Another common approach is to assume a predefined allocation ratio based on water
withdrawal infrastructure (e.g., Siebert et al., 2010), using this ratio to divide total irrigation
abstractions between groundwater and surface water (Arboleda-Obando et al., 2024; Leng et al.,
2015). Despite these advances, the representation of water extraction and the coupling of
irrigation and hydrological systems in LSMs is still in its early stages. Most irrigation-enabled
models still assume an unlimited water supply, failing to account for constraints imposed by
water availability (Druel et al., 2022; Yao et al., 2022; Zhou et al., 2020).
The Common Land Model (CoLM; Dai et al., 2003), derived from the Community Land Model
(CLM), is a widely used land surface model that integrates ecological, hydrological, and
biophysical processes. In recent years, it has further incorporated various physical processes such
as lakes, wetlands, and dynamic vegetation, enhancing the representation of energy and water
exchanges among soil, vegetation, snow, and atmosphere. CoLM has been successfully
implemented in global atmospheric models, such as GRAPES, CWRF, and CAS-ESM2.0 (Shen
et al., 2021; Yuan and Liang, 2011; Zhang et al., 2020a). Despite significant advancements in
parameterizing natural land surface processes, the representation of human activities in CoLM
remains at an early stage. Recently, CoLM has further integrated a crop module, providing a
foundation for considering irrigation and its interactions with natural water systems.
To enhance the representation of human–water interactions in land surface models, we introduce
a new irrigation module for CoLM. This module provides a comprehensive framework for
simulating the entire irrigation water system, including water demand, withdrawal, and
utilization. It incorporates an irrigation water demand scheme based on soil moisture deficits, an
irrigation application scheme accounting for four major irrigation methods, and an irrigation
water withdrawal scheme that incorporates multiple water source constraints by integrating
CoLM with a river routing model and a reservoir operation scheme. A key focus of this module
is the bidirectional coupling between irrigation water demand and supply, alongside a detailed
representation of water withdrawals from different sources. Section 2 provides a detailed



description of the module and its implementation, including an overview of CoLM, the datasets
used for simulation and validation, and the experimental design. Section 3 validates the module's
performance in simulating irrigation water withdrawals using reported data and compares its
results to other hydrological models. It also assesses improvements in model accuracy for
regional energy dynamics (sensible heat, latent heat, and surface temperature), hydrology (river
flow), and agricultural outputs (maize, soybean, and wheat yields). Section 4 demonstrates two
key applications of the module: analyzing irrigation impacts on the energy budget and evaluating
irrigation water security. Finally, we discuss the module's current limitations and propose
potential future improvements.
**2. Materials and Methods**
**2.1 Description of CoLM and its crop module**
The Common Land Model (CoLM) is one of the most advanced land surface models widely used
to simulate the Water–Energy–Carbon Nexus. The original version of CoLM (Dai et al., 2003)
combines the three land surface models: the Land Surface Model (LSM; Bonan, 1996), the
Biosphere-Atmosphere Transfer Scheme (BATS; Dickinson et al., 1993), and the 1994 version of
the Chinese Academy of Sciences Institute of Atmospheric Physics LSM (IAP94; Dai and Zeng,
1997). CoLM2014 integrates the Catchment-Based Macro-Scale Floodplain model (CaMa-
Flood; Yamazaki et al., 2011), enabling river routing calculations within the model. Specifically,
runoff generated by CoLM is transferred to CaMa-Flood for routing through the river network.
CaMa-Flood represents the river network as a series of irregular unit catchments, defined
through sub-grid topographic parameters. River discharge and other flow characteristics are
computed using the local inertial equations along the river network, allowing for detailed flow
dynamics across catchments.
The CoLM2024 version incorporates substantial updates over CoLM2014, particularly by
introducing representations of biogeochemical cycles and human activity processes (e.g., crop
growth and reservoir management). The new crop module introduces a phenological
development scheme based on accumulated temperature, a biomass allocation scheme among
different plant organs, and fertilization schemes (Drewniak et al., 2013). Crops are categorized
into four organ pools: leaves, stems, fine roots, and grains. The growth stages are divided into
three phases: sowing to emergence, emergence to grain filling, and grain filling to maturity, with
carbon allocation ratios to roots, stems, leaves, and grains varying across these phases. Upon
maturation, crops are harvested, with part of the carbon from the grains contributing to the yield,
while a small portion (3g) is reserved as seeds for the next growing season. For carbon
assimilation, the module employs Farquhar's photosynthesis scheme (Collatz et al., 1992;
Farquhar et al., 1980) and Ball-Berry's stomatal model (Ball et al., 1987; Collatz et al., 1991),
treating maize as a C4 crop and other crops as C3. Additionally, the module accounts for the
effects of heat stress, water stress, nitrogen stress, and ozone stress on yield (Li et al., 2024;





Lombardozzi et al., 2020). The module has been calibrated for various crops, including maize,
soybean, spring and winter wheat, rice, cotton, and sugarcane, enabling accurate simulation of
crop yields.

**2.2 Two-way coupled irrigation water use module**

**2.2.1 Irrigation demand**

The irrigation demand is calculated using the soil moisture deficit method (Leng et al., 2017;
Ozdogan et al., 2010; Yao et al., 2022). During the crop growth stage, irrigation is triggered at 6
a.m. local time if the soil moisture in the root zone ($z_{irrig}$=1m) falls below the threshold value
($\omega_{thresh}$). The total irrigation water demand ($D_{irrig}$, mm) is then calculated using Equation (1):
$$D_{irrig} = \begin{cases} \omega_{irrig} - \omega_{avail} & \omega_{avail} \leqslant \omega_{thresh} \\ 0 & \omega_{avail} > \omega_{thresh} \end{cases} \quad (1)$$

where $\omega_{avail}$ is the total soil water amount in the root zone (mm); $\omega_{irrig}$ is the irrigation target
threshold (mm), calculated using Equation (2):
$$\omega_{irrig} = f_{irrig}(\omega_{target} - \omega_{wilt}) + \omega_{wilt} \quad (2)$$

where $\omega_{wilt}$ is the wilting point soil water amount in the root zone (mm), calculated as the sum of
soil water at the wilting point for each soil layer ($\sum_{j=1}^{N_{irr}} \theta_{wilt}\Delta z_j$); $\omega_{target}$ is the target soil water
amount in the root zone (mm), calculated as the sum of target soil water for each soil layer
($\sum_{j=1}^{N_{irr}} \theta_{target}\Delta z_j$). $N_{irr}$ is the number of soil layers in the root zone and $\Delta z_j$ is the thickness of each
soil layer (m). The target ($\theta_{target}$) and wilting point ($\theta_{wilt}$) soil moisture (m³/m³) for each layer are
calculated based on the corresponding soil water potential ($\Phi_{target}$ and $\Phi_{wilt}$). $f_{irrig}$ is a weighting
coefficient ranging from 0 to 1, controlling the extent to which soil water amount approaches the
target level $\omega_{target}$ during irrigation (default value = 1). In some cases, it can represent the
efficiency of the irrigation system, accounting for water losses due to evaporation, seepage, or
other factors.
The irrigation trigger threshold ($\omega_{thresh}$) in Equation (1) is calculated as:
$$\omega_{thresh} = f_{thresh}(\omega_{trigger} - \omega_{wilt}) + \omega_{wilt} \quad (3)$$

where $\omega_{trigger}$ is the trigger water amount in the root zone (mm); $f_{thresh}$ is also a weighting
coefficient ranging from 0 to 1 that controls the proximity of soil water amount to the trigger
level $\omega_{trigger}$ (default value = 1).





The values of $\omega_{\text{trigger}}$ and $\omega_{\text{target}}$ are set according to the irrigation application method. For drip
and sprinkler irrigation, both $\omega_{\text{trigger}}$ and $\omega_{\text{target}}$ are set to the soil field capacity water amount. For
flood irrigation, $\omega_{\text{trigger}}$ is set to the soil field capacity water amount and $\omega_{\text{target}}$ to the saturation
water amount. For paddy irrigation, both $\omega_{\text{trigger}}$ and $\omega_{\text{target}}$ are set to the saturation water amount.
**2.2.2 Irrigation application**
The model incorporates four different irrigation application methods: drip irrigation, sprinkler
irrigation, flood irrigation, and paddy irrigation, each with unique triggering conditions, water
demand requirements, and application processes. Drip irrigation is triggered when soil moisture
in the root zone falls below field capacity, with the irrigation goal being to restore soil moisture
to field capacity. This method applies water directly to the surface soil, allowing it to percolate
into deeper soil layers. Sprinkler irrigation shares the same triggering condition and demand
requirement as drip irrigation but applies water above the canopy. In this method, water can be
intercepted and evaporated before reaching the soil surface, resulting in relatively lower
irrigation efficiency. This method is the most commonly used in the United States. Flood
irrigation is triggered when soil moisture falls below field capacity, to raise soil moisture to the
point of saturation. Paddy irrigation is applied whenever soil moisture drops below saturation,
aiming to restore soil moisture to saturation without causing runoff loss. Paddy fields are
typically maintained with a specific water level on the surface (10 cm) during the growing
season. A global irrigation method map (Yao et al., 2022; Figure S3) is used to determine the
irrigation method for each grid. In addition, irrigation is implemented daily at 6 a.m., if
necessary, with water supply evenly distributed across each time step throughout the next 4
hours.
**2.2.3 Irrigation water supply/withdrawal**
The model incorporates two distinct irrigation water supply/withdrawal schemes. The first
scheme, Unlimited Supply (irrig-unlim), assumes that irrigation demand is fully met without
accounting for specific water sources, a common approach in most land surface models (Yao et
al., 2022). The second scheme, Limited Supply (irrig-lim), divides total irrigation demand
between surface water and groundwater sources, labeled as surface water demand ($D_{\text{surf}}$) and
groundwater demand ($D_{\text{grnd}}$), respectively. Both demands are constrained by the available water
within each respective system. This distribution is based on the spatial extent of groundwater
irrigation equipment, as provided by Siebert et al. (2010), and is formulated as follows:

$$D_{\text{surf}} = D_{\text{irrig}} \times (1 - f_{\text{grnd}}) \tag{4}$$

$$D_{\text{grnd}} = D_{\text{irrig}} \times f_{\text{grnd}} \tag{5}$$



where $D_{surf}$ and $D_{grnd}$ represent the demand from surface water and groundwater systems, and
$f_{grnd}$ denotes the area fraction covered by groundwater equipment. In this scheme, surface water
demand ($D_{surf}$) is sourced sequentially from local grid cell runoff, local river streamflow, and
upstream reservoirs, while groundwater demand ($D_{grnd}$) is drawn from groundwater aquifers.
**2.2.3.1 Surface water supply**
In our two-way coupled irrigation system (Figure 1), the daily surface water supply for irrigation
is constrained by surface water availability, which is simulated by CoLM (runoff) and CaMa-
Flood (local streamflow and upstream reservoirs). We first examine whether the runoff from the
local grid cell ($S_{ro}$) can meet the daily surface water demand ($D_{surf}$) for that cell. If runoff is
insufficient, additional water is sourced from local streamflow and upstream reservoirs. River
streamflow availability ($S_{riv}$) is determined by CaMa-Flood. For each irrigated grid cell, the river
grid with the highest flow within a 250 km radius is selected as the source. To prevent excessive
water extraction, a withdrawal limit is imposed, ensuring that the remaining flow in each river
grid cell does not drop below 20% of its average daily volume. Before conducting irrigation
simulations, natural river flow simulations are performed to establish essential parameters for
both river and reservoir water withdrawal schemes.
Reservoir water availability ($S_{res}$) is also determined by CaMa-Flood, which now includes a
reservoir module. This module consists of the following components: (i) a reservoir dataset that
provides reservoir location information matched with the river network, along with reservoir
parameters (e.g., characteristic storage capacity); (ii) a reservoir operation scheme designed for
flood control; and (iii) a routing scheme that integrates reservoir operations into river flow
simulations. For more details, refer to Hanazaki et al. (2022). In this study, we further propose a
new scheme for sourcing irrigation water from reservoirs (Figure 2), which involves the
following steps:
(i) Identifying the irrigation area served by each reservoir. It is challenging to accurately define
the true irrigation extent/area for each reservoir, especially across large spatial domains.
Therefore, a simplified approach is adopted here: larger reservoirs are assumed to cover a
proportionately larger irrigation area, restricted to downstream regions only (since upstream
water transfer is economically infeasible). Based on the relationship between reservoir size and
the corresponding irrigation area provided in Table S1, we calculate the irrigation area for each
reservoir according to its storage capacity by linear interpolation. Downstream irrigation grids
are selected sequentially, from nearest to farthest, until the cumulative grid area closely matches
the calculated irrigation area. If multiple reservoirs serve the same irrigation grid, a sharing
proportion ($f_{share}$, ranging from 0 to 1) is assigned to the irrigation grid based on the degree of
shared usage.



(ii) Calculating the irrigation demand for each reservoir by aggregating the demands of
associated irrigation grids. This is expressed as: $D_{\text{res-total}} = \sum_{i=1}^{N} \left( D_{\text{irrig-unmet}}^{i} \times f_{\text{share}}^{i} \right)$, where
$D_{\text{irrig-unmet}}^{i}$ and $f_{\text{share}}^{i}$ represent the irrigation demand (i.e., the portion of $D_{\text{surf}}$ not met by local
runoff and river streamflow) and sharing proportion of grid $i$, respectively. $N$ denotes the number
of irrigation grids served by the reservoir.
(iii) Executing reservoir withdrawals for irrigation based on demands. Water is then withdrawn
($S_{\text{res-total}}$) from the reservoir's effective storage ($V_{\text{eff}}$) — the portion between the current water
level and dead water level—according to the required demand. This is expressed as
$S_{\text{res-total}} = \min(D_{\text{res-total}}, V_{\text{eff}})$. After updating the reservoir storage, the reservoir operation and
subsequent river routing are calculated following the approach outlined in Hanazaki et al. (2022).
(iv) Redistributing withdrawn water to the irrigation grids. Based on each irrigation grid's
contribution to the total reservoir irrigation demand, the total withdrawal volume is
proportionally allocated across the associated grids ($S_{\text{res}}^{i}$). This is expressed as
$S_{\text{res}}^{i} = S_{\text{res-total}} \times \frac{D_{\text{irrig-unmet}}^{i} \times f_{\text{share}}^{i}}{D_{\text{res-total}}}$. Notably, this water is not applied directly to irrigation but is stored in
a temporary reservoir (i.e., a temporary variable) for each irrigation grid in CoLM. This approach
addresses the response delay in water supply from the river routing model to the land model's
irrigation demands, as the time step for CoLM is 60 minutes, while CaMa-Flood operates with a
6-hour time step and exchanges information with CoLM every 6 hours. Moreover, if the
reservoir cannot fully meet the irrigation demand within the initial time step, any unmet demand
is carried forward to the next time step. This process continues over a 24-hour cycle, after which
new water demands for the next day are received.
Thus, the computational sequence proceeds as follows: Step (i) is completed before model
execution, with its results serving as an essential input for the irrigation module. During model
operation, CoLM calculates the irrigation demand at 6 a.m. local time. The unmet demand (after
subtracting the water supplied by local runoff and streamflow) is then sent to CaMa-Flood, as
described in Step (ii). CaMa-Flood supplies water from reservoir to meet this demand, as
described in Step (iii), and returns the supplied water to CoLM according to Step (iv), over the
next 24 hours. During this process, the water supplied by reservoir is stored in the temporary
reservoir (variable) for each irrigation grid within CoLM. The following day, when irrigation
begins again at 6 a.m., water is withdrawn directly from the temporary reservoir if the demand
cannot be met by local runoff and streamflow.





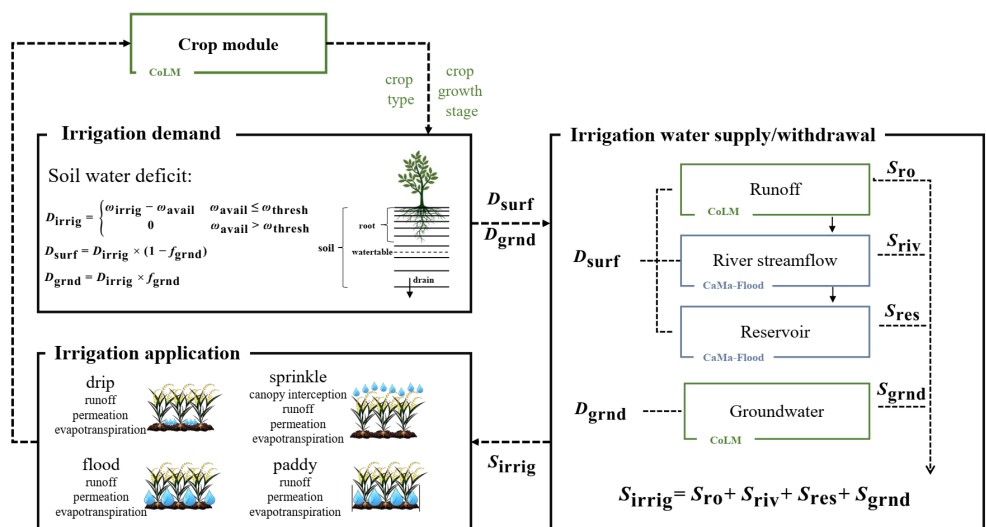

**Figure 1.** Diagram of the two-way coupled irrigation water system in the Common Land Model.

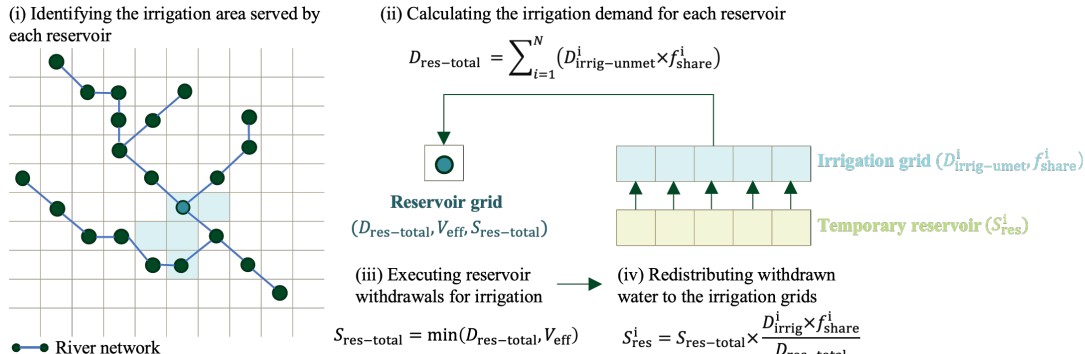

**Figure 2.** Diagram of the reservoir water supply scheme.
**2.2.3.1 Groundwater supply**
Groundwater supply is constrained by the availability of water within the aquifer. In CoLM, the
groundwater table interacts with soil layers through vertical water exchange, allowing recharge
or withdrawal of water from the aquifer (Li et al., 2017). The evolution of the groundwater table
is determined by the balance of soil water recharge and subsurface outflow, with the specific
yield dynamically linking the water table position to changes in soil moisture and aquifer
storage. When irrigation is required, water is directly extracted from the top of the simulated
aquifer, and the water table depth is updated accordingly. This process continues until either the
irrigation demand is fully met, or the water table falls below a predefined threshold, set as 1
meter below the initial depth at the beginning of the year (Jasechko et al., 2024; Russo and Lall,
2017). Groundwater supply is immediately available upon demand, with no temporal lag



between the request and its availability for irrigation. Changes in the water table depth can then
affect subsurface drainage and recharge from the bottom soil layer to the aquifer.
**2.3 Materials**
**2.3.1 Input datasets**
In this study, CoLM was implemented across the contiguous United States at a 0.25° spatial
resolution for the period 2001–2016. Meteorological input data were derived from the WATCH
Forcing Data methodology applied to ERA-Interim data (WFDEI) (Weedon et al., 2014), which
has also been utilized in the Inter-Sectoral Impact Model Intercomparison Project Phase 2a
(ISIMIP2a; Gosling et al., 2019). Soil property data were sourced from the Global Soil Dataset
for Earth System Modeling (GSDE), originally provided at a spatial resolution of 30 arc-seconds
(Dai et al., 2019; Shangguan et al., 2014). Land cover data were derived from the MODIS
dataset (MCD12Q1; Friedl and Sulla-Menashe, 2022), providing detailed global land
classification information at a spatial resolution of 500 meters.
The simulation of irrigation processes also required detailed data on crop areas, planting dates,
irrigation areas and irrigation methods. Crop planting areas were derived from the 30-meter
resolution CropScape and Cropland Data Layer (CDL) datasets (2008–2020) and aggregated to a
spatial resolution of 5 arcminutes for analysis (USDA, 2019). These datasets, produced by the
U.S. Department of Agriculture, provide annual, crop-specific land cover information using
satellite imagery and ground reference data. For each pixel, we calculated the proportion of
cropland relative to the pixel's area (PCT_CROP) and the proportions of maize, wheat, and
soybean relative to the cropland area (PCT_CFT). Pixels with a cropland percentage
(PCT_CROP) exceeding zero were classified as crop pixels. The Plant Functional Type (PFT)
approach employed in CoLM allowed different crops and vegetation types coexist within the
same grid cell according to their percentages (PCT_CFT). To define planting and harvesting
dates, we utilized an observation-based crop calendar dataset from the Global Gridded Crop
Model Intercomparison (GGCMI), which provided information for 20 major crops under both
rainfed and irrigated conditions at each grid cell for 1980–2010 (Jägermeyr et al., 2021).
The irrigation map was derived using the 5' resolution data from the FAO Global Map of
Irrigation Areas - Version 5 (Siebert et al., 2013). Since the CropScape data does not distinguish
between rainfed and irrigated crops, we combined it with the irrigation map to determine the
proportions of rainfed and irrigated crops. Irrigation water withdrawals were classified into
surface water and groundwater sources following FAO data on regions equipped for groundwater
extraction, which informed the allocation of irrigation demand across sources (Siebert et al.,
2010). The irrigation application method data was obtained from Yao's global irrigation map,
which details irrigation methods (drip, sprinkler, or flood) for 32 crop types, each assigned a
single method (Yao et al., 2022). Jägermeyr et al. (2015) originally used a decision tree to refine



AQUASTAT's data, classifying irrigation methods for 14 Crop Functional Types (CFTs) based
on crop area, soil characteristics, and socio-economic conditions. Yao et al. (2022) then matched
these CFTs to 32 crop types in CLM5 and incorporated an additional irrigation method, *paddy*,
specifically for rice-growing regions, creating a more detailed global irrigation dataset.
For river routing simulations in CaMa-Flood, the baseline topography was derived from the
Multi-Error-Removed Improved-Terrain Hydrography dataset (MERIT Hydro; Yamazaki et al.,
2019). Fundamental information on dams/reservoirs in the river network, including dam name,
coordinates, total storage capacity, and drainage area, was obtained from the GRanD database
(Lehner et al., 2011). GRanD version 1.3 contains data on 7,320 dams globally, along with their
associated reservoirs. The locations of the dams in the 0.25° river map were determined
following the method outlined by Hanazaki et al. (2022), which enabled the identification of
1464 reservoirs across the contiguous United States (Figure S2). In addition to GRanD, the
Global Reservoir Surface Area Data Set (GRSAD; Zhao and Gao, 2018) and the Global
Reservoir Geometry Database (ReGeom; Yigzaw et al., 2018) were used to estimate reservoir
parameters, such as storage capacity at emergency, flood control, and critical levels (Hanazaki et
al., 2022). GRSAD provides a monthly time series of surface areas for 6,817 GRanD reservoirs
from 1984 to 2015, based on global surface water occurrence data (Pekel et al., 2016). ReGeom
contains storage-area-depth information for 6,824 reservoirs in GRanD, with geometry estimates
derived from assumed surface and cross-sectional shapes, as well as data on reservoir extent,
total storage, and surface area.
**2.3.2 Validation datasets**
To evaluate the scheme developed in this study, we focused on validating irrigation water
withdrawal volumes, land fluxes (including energy fluxes and river flows) and crop yields in
irrigated areas. We used hydrological survey data from the U.S. Geological Survey (USGS,
2023), which provided detailed statistics on total irrigation water withdrawals, categorized by
surface and groundwater sources, every five years since 2000. Within the timeframe of this
study, data were available for the years 2005, 2010, and 2015. Building on this, Ruess et al.
(2024) employed a global hydrological model (PCR-GLOBWB) to estimate annual, crop-
specific irrigation water withdrawals from 2008 to 2020. Additionally, we compared the
irrigation water withdrawal volumes simulated by our model with those generated by six other
hydrological models—VIC, PCR-GLOBWB, MATSIRO, LPJmL, H08, and DBH—that
participated in ISIMIP2a (Gosling et al., 2019). Although more hydrological models were
included in ISIMIP2a, our comparison was limited to these six because they provided irrigation
water withdrawal outputs. The simulations were driven by the WFDEI climate dataset, with a
spatial resolution of 0.5° and covering the period from 1971 to 2010.
For land surface flux validation, we used monthly latent and sensible heat fluxes provided by
FLUXCOM at a resolution of 0.5° (Jung et al., 2019). FLUXCOM leveraged FLUXNET site





observations and extended these globally through machine learning algorithms, resulting in a
global dataset for latent heat, sensible heat, and carbon fluxes. For temperature validation, we
used land surface temperature data from 2001 to 2016 at a spatial resolution of 0.1° from the
ERA5-Land reanalysis dataset (Muñoz-Sabater et al., 2021).
For streamflow validation, we utilized monthly streamflow data from the Global Runoff Data
Centre (GRDC, 2023) for the period 2001–2016. To ensure robust validation, we excluded
catchments with fewer than five years of data during the study period and focused on catchments
significantly influenced by irrigation while minimizing the impacts of other anthropogenic
activities. These selection criteria ultimately resulted in 77 catchments being included in the
analysis (Figure S7).
For crop yield validation, we relied on annual yield reports for irrigated and rainfed crops from
the USDA NASS at the county level, which is regarded as a reliable source of yield statistics
(USDA/NASS, 2023). The data for irrigated crops primarily covered the Central Plains of the
United States, with limited coverage in the eastern and western regions. We aggregated our grid-
based yield simulation results to the county level and performed validation only for regions and
years with available USDA data.
**2.4 Experimental Design**
This study conducted three simulation experiments to evaluate the effectiveness of the newly
developed module by comparing their performance: (i) Non-Irrigation Experiment (abbreviated
as noirrig): This scenario assumes all crops in the region are rainfed, with no irrigation applied. It
serves as a baseline to represent natural surface water and energy balance conditions. (ii)
Unlimited Irrigation Experiment (abbreviated as noirrig-unlim): This scenario distinguishes
between irrigated and rainfed areas based on crop maps. In irrigated areas, crop water demands
are fully satisfied throughout the growing season, without considering the limitations of water
resources. (iii) Limited Irrigation Experiment (abbreviated as irrig-lim): In this scenario,
irrigation water is supplied proportionally from surface water and groundwater based on
availability, as illustrated in Figure 1. Here, irrigation is constrained by the availability of surface
and groundwater, which may result in unmet crop water demands.
The non-irrigation experiment was first simulated for the period 2001–2010 to stabilize
vegetation carbon and nitrogen pools, soil moisture, and the groundwater table. This stabilized
state served as the initial condition for all three experiments. The main simulation period spanned
2001-2016, covering the contiguous United States at a spatial resolution of 0.25° × 0.25°. In the
subsequent analysis, key evaluation metrics—bias, root-mean-square error (RMSE), Pearson
correlation coefficient (*r*), and Kling-Gupta efficiency (KGE)—were employed to assess the
performance of the simulations.





**3. Results**
**3.1 Evaluation of simulated irrigation water withdrawal**
**3.1.1 Comparison with observations**
Based on annual irrigation withdrawal data from the USGS, states in the western and central
United States withdraw significantly more water for irrigation than those in the eastern regions
(Figure 3a). This is primarily due to the relatively low precipitation in the western and central
regions, where the majority of irrigated areas are located, while crops in the eastern U.S. are
predominantly rainfed. The top five states with the highest annual irrigation withdrawals—
California (CA), Idaho (ID), Colorado (CO), Arkansas (AR), and Montana (MT)—are all
situated in the Midwest and West (Figure 3b). Nationally, the total annual irrigation withdrawal
averages approximately 166.23 $km^3 yr^{-1}$, based on data from 2005, 2010 and 2015. In
comparison, the irrig-unlim and irrig-lim schemes simulate national total withdrawals of 290.94
$km^3 yr^{-1}$ and 120.81 $km^3 yr^{-1}$, respectively. As illustrated in Figure 3c-f, the simulations capture
the spatial patterns of water withdrawals across different states effectively, with the irrig-lim
scheme yielding better performance. The root-mean-square-error (RMSE) and correlation
coefficient ($r$) for the irrig-lim scheme are 3.60 $km^3 yr^{-1}$ and 0.82, respectively, slightly
outperforming the corresponding values for the irrig-unlim scheme (9.78 $km^3 yr^{-1}$ and 0.76).
Irrigation water withdrawals draw from both surface water and groundwater sources. According
to USGS reports, most irrigation withdrawals in the central U.S. come from groundwater (Dieter
et al., 2018). In states such as Missouri (MO), Kansas (KS), Iowa (IA), Illinois (IL), Rhode
Island (RI), and Mississippi (MS), the share of groundwater withdrawals exceeds 90% (Figure
4c). In contrast, states with high surface water withdrawals are primarily in the eastern and
western U.S., with states like Wyoming (WY), Connecticut (CT), Kentucky (KY), and Montana
(MT) reporting surface water withdrawal proportions greater than 90%. These spatial variations
in water source usage are primarily attributed to the central U.S.'s abundant groundwater
resources and widespread groundwater extraction infrastructure.
In our simulations, the irrig-lim scheme effectively accounts for irrigation water withdrawals
from different sources, constrained by their availability. Encouragingly, the scheme generally
reproduces observed annual surface water and groundwater withdrawals across states (Figure 4a-
b), achieving correlation coefficients of 0.68 and 0.95, respectively. Furthermore, the simulated
proportions of water sources closely align with observed data (Figure 4c-d), with a correlation
coefficient of 0.64 ($p < 0.01$). However, the model tends to underestimate the surface water
withdrawal proportions in the northwestern regions of the U.S. (particularly in Montana and
Colorado; Figure 4d), while slightly overestimating them in some central and eastern states. This
discrepancy may stem from limitations in the data used to allocate water demand. Specifically,
the model relies on pre-determined groundwater extraction infrastructure proportions, which may



not accurately reflect actual extraction practices, particularly as the dataset was published in
2005 and may not account for subsequent changes in groundwater infrastructure in certain states.
Alternatively, the discrepancy could arise from model biases in simulating surface and
groundwater availability. For example, in the northwestern region, surface runoff is heavily
influenced by snowmelt and glacial meltwater (Li et al., 2017), and biases in simulating these
processes could lead to an underestimation of surface water availability.

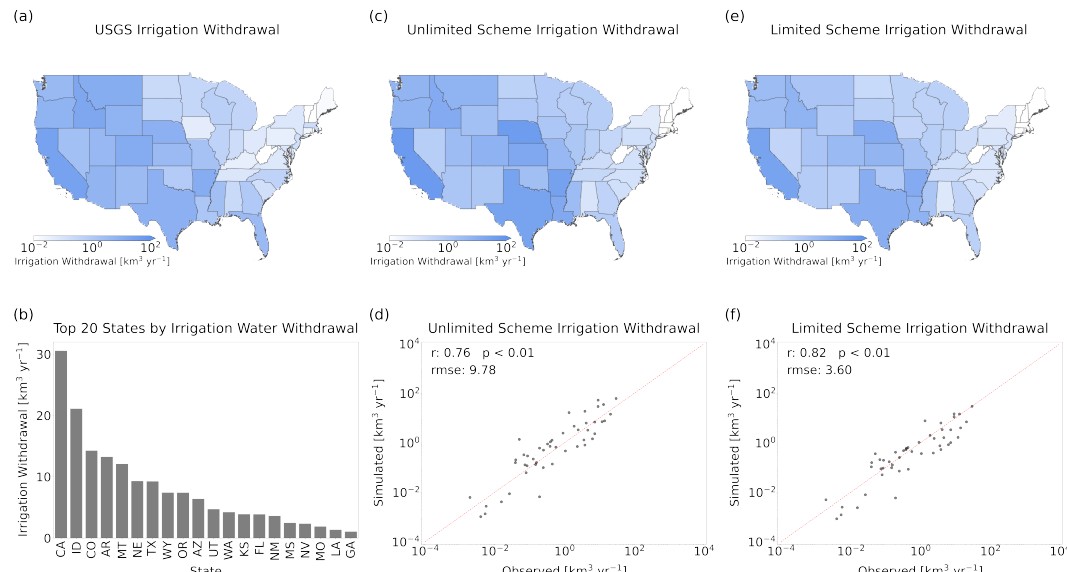

**Figure 3.** Comparison of reported and simulated irrigation water withdrawal in the United States.
(a) Annual irrigation water withdrawal reported by the USGS for individual states. (b) Annual
withdrawal amounts for the top 20 states by irrigation water withdrawal. (c) Annual irrigation
water withdrawal simulated by CoLM using the unrestricted water supply (irrig-unlim) scheme
for individual states. (d) Comparison of reported and simulated irrigation water withdrawal
(using the irrig-unlim scheme) for individual states, with Pearson correlation coefficient ($r$) and
root mean square error (RMSE) displayed, along with statistical significance (two-tailed
Student's t-test). (e) Annual irrigation water withdrawal simulated by CoLM using the restricted
water supply (irrig-lim) scheme for individual states. (f) Comparison of reported and simulated
irrigation water withdrawal (using the irrig-lim scheme) for individual states.



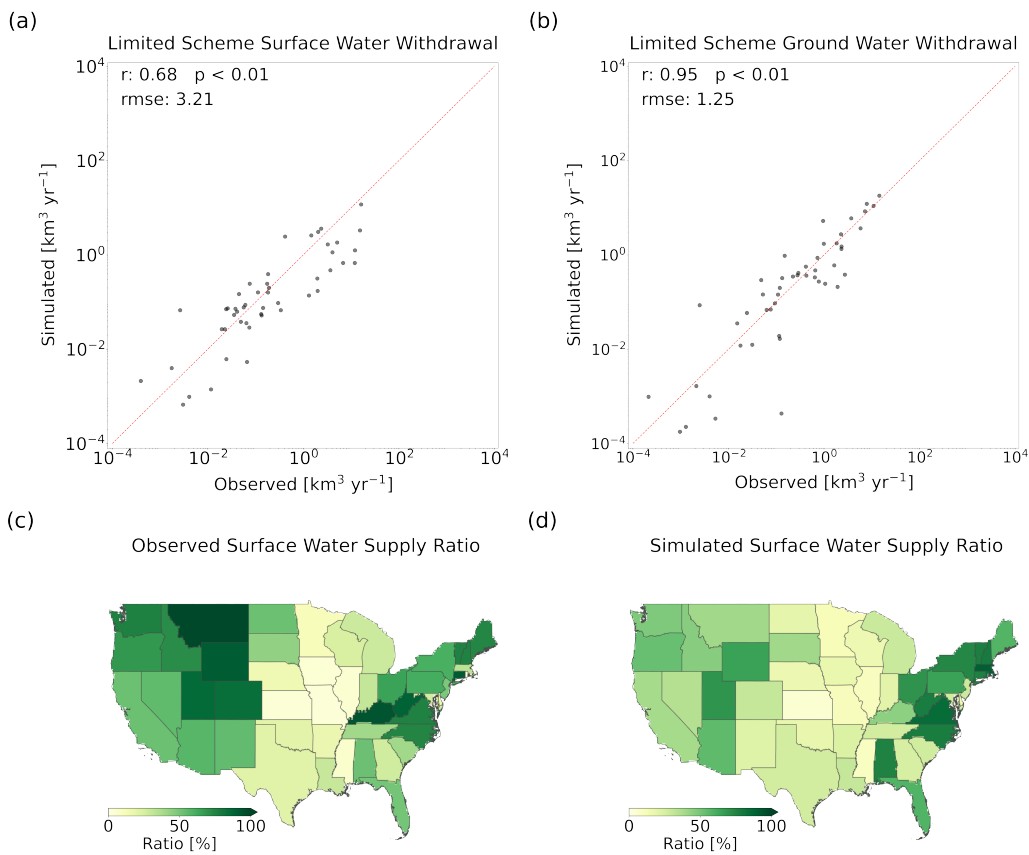

**Figure 4.** Comparison of reported and simulated irrigation water withdrawal in the United States by water source. (a) Comparison of reported and simulated surface water withdrawal volumes for individual states. (b) Same as (a), but for groundwater withdrawal volumes. (c) Proportion of surface water in irrigation withdrawal, based on USGS reports for individual states. (d) Proportion of surface water in irrigation withdrawal, simulated by CoLM using the irrig-lim scheme for individual states.

Ruess et al. (2024), using data from the USGS and model outputs from PCR-GLOBWB 2, generated an irrigation water withdrawal dataset that included withdrawal volumes for major crops in the U.S. According to this dataset (Figure 5), wheat is the largest consumer of irrigation water, with an average annual withdrawal of approximately 27.29 km$^3$ yr$^{-1}$, followed by maize at about 20.91 km$^3$ yr$^{-1}$. In contrast, soybean requires considerably less irrigation (i.e., 5.89 km$^3$ yr$^{-1}$), partly due to its greater drought tolerance and smaller planted area compared to the other two crops. Under the irrig-unlim (irrig-lim) simulation scheme, the annual irrigation withdrawals for maize, wheat, and soybean are 53.98 km$^3$ yr$^{-1}$, 47.53 km$^3$ yr$^{-1}$, and 29.99 km$^3$ yr$^{-1}$ (19.19 km$^3$ yr$^{-1}$,





17.95 km$^3$ yr$^{-1}$, and 11.05 km$^3$ yr$^{-1}$), respectively. Once again, the irrig-lim scheme provides a
closer alignment with observation-based data, as indicated by a lower RMSE (Figure 5). These
results suggest that our irrigation module generally performs well in simulating total national
annual water withdrawals, the spatial distribution of withdrawals (Figure 3), the proportion of
water source types (Figure 4), and the irrigation volumes for different crops (Figure 5).

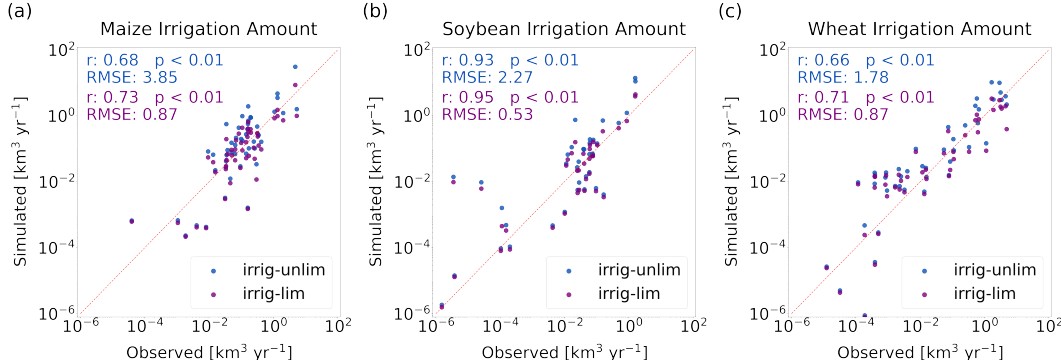

**Figure 5.** Comparison of reported and simulated irrigation water withdrawal in the United States
by crop type. (a) Comparison of reported and simulated irrigation water withdrawal for maize,
using both the unrestricted (irrig-unlim, blue dots) and restricted (irrig-lim, purple dots) supply
schemes for individual states. (b-c) Same as (a) but for soybean and wheat.
**3.1.2 Comparison with other models**
We further compare the irrigation water withdrawal simulations from this study with outcomes
from six global hydrological models (VIC, PCR-GLOBWB, MATSIRO, LPJmL, H08, and
DBH) that participated in ISIMIP2a. Notably, all simulations used the same climate forcing
(WFDEI), ensuring consistency in the comparison. Our results, particularly from the irrig-lim
scheme, closely align with observed total national annual irrigation withdrawals. By contrast,
five of the six models, excluding LPJmL, exhibit larger absolute deviations from observed value
(Figure 6a). Regarding spatial distribution, most models perform well (Figure 6b), with LPJmL
(orange dots) achieving the highest correlation coefficient (0.89) and the lowest RMSE (2.86 km$^3$
yr$^{-1}$). The irrig-lim scheme in this study (purple dots) performs comparably to LPJmL,
demonstrating competitive accuracy. In terms of temporal dynamics, comparisons across models
are limited due to the scarcity of observed data. However, the general seasonal patterns are
consistent across models (Figure S5), with the highest irrigation withdrawals occurring in June
and July, and the lowest in January and December. Most models exhibit similar seasonal
fluctuations, with irrigation volumes during peak months approximately ten times greater than
during off-peak months. Overall, these results suggest that our model performs similarly to, or
even better than, existing models in simulating irrigation water withdrawals in the U.S.




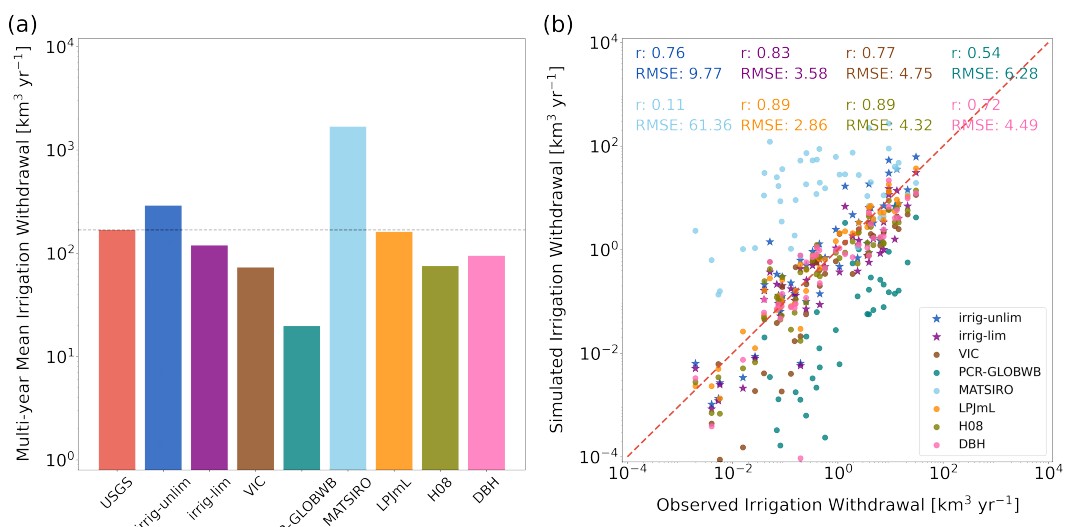


**Figure 6.** Comparison of irrigation water withdrawal simulated by CoLM and six global
hydrological models participating in ISIMIP2a. (a) Annual total irrigation water withdrawal
amounts in the United States as reported by the USGS, compared with simulations from CoLM
(using both the irrig-unlim and irrig-lim schemes) and the six global hydrological models. (b)
Comparison of reported and simulated irrigation water withdrawal for individual states, with
Pearson correlation coefficient (*r*) and root mean square error (RMSE) for each simulation
displayed.

**3.2 Evaluation of simulated land energy and water fluxes**

**3.2.1 Evaluation of simulated energy fluxes**

We evaluate CoLM's performance in simulating surface energy fluxes over irrigated areas in the
U.S. using different schemes, with FLUXCOM monthly sensible heat (*SH*) and latent heat (*LH*)
fluxes as observational references. Figure 7 compares multi-year monthly averages of observed
and simulated *SH* and *LH* fluxes across irrigated grid points. Without irrigation (the noirrig
scheme), the model significantly overestimates *SH* (Figure 7a) with an average bias of 16.89 W
m$^{-2}$ (44.53%) and underestimates *LH* (Figure 7c) with an average bias of 18.84 W m$^{-2}$ (37.11%).
In contrast, biases over non-irrigated grids are considerably lower, at 3.04% and 17.38% for *SH*
and *LH*, respectively (Figure S6). This indicates that CoLM performs satisfactorily in simulating
energy processes over natural vegetation and rainfed areas, but less so over irrigated regions.

Upon introducing the irrigation module, the simulation errors in surface energy fluxes over
irrigated areas are significantly reduced. Under the irrig-unlim and irrig-lim schemes, average *SH*
biases decrease to 27.10% and 30.79%, respectively, while *LH* biases decrease to 18.41% and





22.18%. These improvements are evident across most irrigated grid points, as illustrated by the
KDE curves of KGE, which indicate an increase in grid points with higher KGE values (Figure
7). A KS test confirms that the differences between the irrigation (i.e., the irrig-unlim and irrig-
lim schemes) and noirrig simulations are statistically significant. Although the irrig-unlim
scheme performs slightly better than irrig-lim for *SH* and *LH*, this difference is not significant.

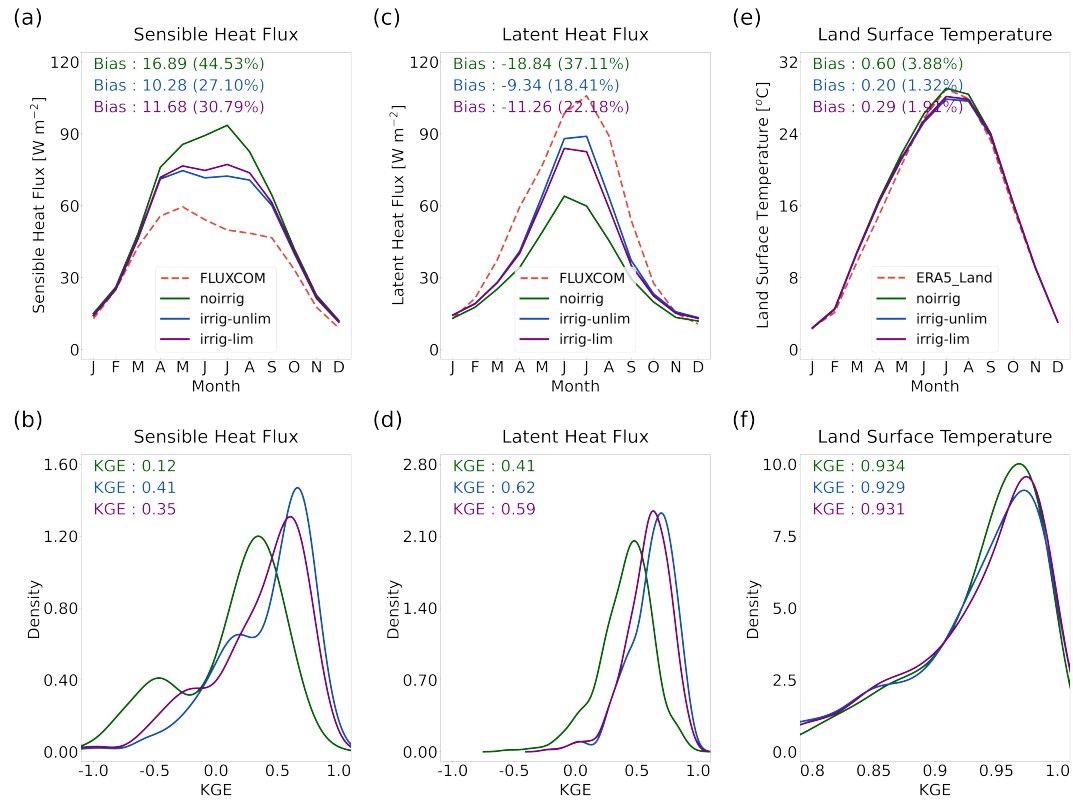

**Figure 7.** Evaluation of simulated energy fluxes and land surface temperature in the irrigation
region. (a) Monthly sensible heat flux averaged from 2001 to 2016, based on FLUXCOM dataset
and simulated by CoLM using the noirrig, irrig-unlim, and irrig-lim schemes in irrigation regions
of the United States, with the bias between simulations and observations (i.e., FLUXCOM)
indicated in the panel. (b) Kernel density estimate (KDE) curves for the Kling-Gupta efficiency
(KGE) between observed and simulated monthly sensible heat flux for each irrigation grid, with
mean KGE value indicated in the panel. (c-d) Same as (a-b) but for latent heat flux. (e-f) Same as
(a-b) but for land surface temperature, using data from ERA5-Land reanalysis dataset.
Additionally, the FLUXCOM data (red dashed line) show that the highest monthly *SH* and *LH*
occur in May and July, respectively. However, the noirrig simulation (green solid line) fails to
capture this seasonal peak, showing instead that *SH* peaks in July and *LH* in June. This





discrepancy is not present in non-irrigated areas (Figure S6), suggesting that irrigation in
agricultural regions (and the subsequent crop growth it supports) substantially affects the
seasonal pattern of regional energy balance. When the irrigation module is incorporated into the
model, these seasonal patterns are more accurately reproduced, with the timing of the simulated
peak months aligning more closely with FLUXCOM data (blue and purple solid lines).
The incorporation of the irrigation module improves the simulation of energy partitioning in
irrigated areas, enabling the model to better capture surface temperature dynamics (Figure 7e).
Under the noirrig scheme, the average bias of monthly surface temperature is 0.6℃ (3.88%).
This bias decreases to 0.20℃ (1.32%) with the irrig-unlim scheme and 0.29℃ (1.91%) with the
irrig-lim scheme. However, even with irrigation included, the simulated total evapotranspiration
remains systematically underestimated (Figure 7c). This underestimation is also evident in non-
crop areas (Figure S6c), suggesting that it may not be due to limitations in the irrigation module
itself but rather to certain deficiencies in CoLM's evapotranspiration simulation approach.
**3.2.2 Evaluation of simulated river flow**
Irrigation processes can significantly alter natural hydrological dynamics and river flow patterns
both temporally and spatially. To evaluate the effectiveness of the irrigation module in capturing
these impacts, we compare model outputs with observed catchment streamflow data. We select
catchments that are substantially influenced by irrigation while minimizing the effects of other
anthropogenic activities. Figure S7 illustrates the locations of the selected 77 catchments. Figure
8 shows that CoLM's performance in simulating runoff—and consequently streamflow—
remains limited, with relatively low average KGE values across all three schemes. This
limitation is likely due to the use of a simplified runoff parameterization scheme in CoLM (Li et
al., 2011). However, it is encouraging to note that the irrig-lim scheme notably improves monthly
streamflow simulations compared to the noirrig scheme, increasing the average KGE from -0.57
to -0.49 and reducing the average percentage bias (PBIAS) from 117.28% to 106.54%. The
enhancement can be largely attributed to the incorporation of irrigation effects, which account
for reduced streamflow due to increased water use for evapotranspiration. This adjustment
effectively mitigates the overestimation of streamflow observed in the noirrig scheme.
Furthermore, our analysis reveals that the irrig-unlim scheme significantly reduces the accuracy
of streamflow simulations compared to the noirrig scheme, leading to a pronounced
overestimation of river discharge. The average relative bias increases substantially from 117.28%
to 147.23% (Figure 8b). This issue arises because the irrig-unlim scheme meets any irrigation
demand by introducing additional water directly into the system without considering its source.
Such an approach is common among crop and land surface models that incorporate irrigation
(Malek et al., 2017; Yao et al., 2022; Zhang et al., 2020b). However, our findings indicate that
introducing extra water for irrigation without accounting for its specific sources and limitations
may lead to an imbalance in the water budget from a comprehensive perspective of the entire





water system, undermining the model's ability to accurately represent the dynamics of the
hydrological system.

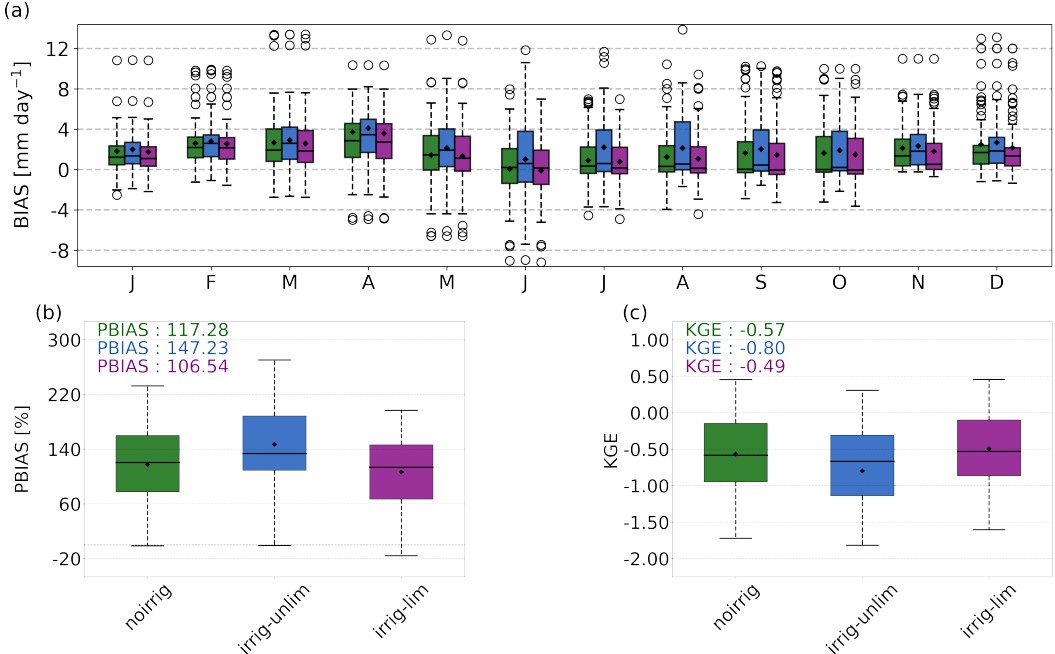


**Figure 8.** Evaluation of simulated streamflow in 77 irrigation-affected catchments. (a) Multi-
year average monthly streamflow bias simulated using the noirrig, irrig-unlim, and irrig-lim
schemes in the evaluation catchments. The boxes represent the interquartile range, black lines
indicate median values, black dots show mean values, and dashed black whiskers extend to 1.5
times the interquartile range; points outside the boxes represent outliers. (b) Percentage bias
(PBIAS) between observed monthly streamflow and simulations from CoLM under the noirrig,
irrig-unlim, and irrig-lim schemes, with the average PBIAS value indicated in the panel. (c)
Same as (b) but for the Kling-Gupta efficiency (KGE) between simulated and observed
streamflow.
**3.3 Evaluation of simulated crop yield**
Irrigation reflects a direct human influence on crop yields by providing supplemental water. Crop
models primarily focus on this aspect, but they often neglect how irrigation affects other
processes. Conversely, most hydrological models concentrate on the impact of irrigation
withdrawals on the water cycle, with some also addressing energy fluxes, yet pay less attention
to crop yield. From this perspective, land surface models offer distinct advantages; they provide
a more detailed representation of hydrological and surface energy processes compared to crop





models, while also presenting more physics-based representations of crop growth than traditional
hydrological models. Therefore, this study further evaluates whether incorporating the developed
irrigation module can enhance crop yield the simulations.

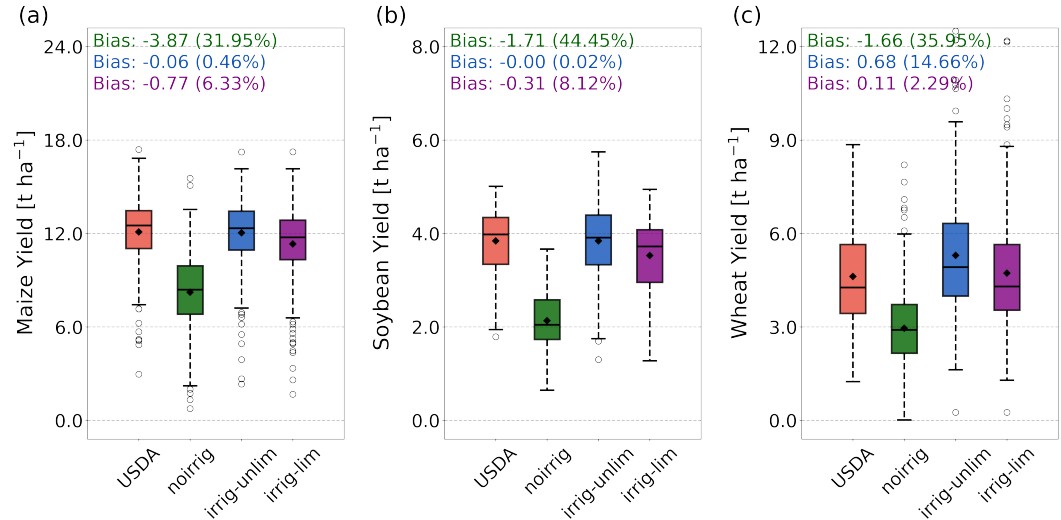

**Figure 9.** Evaluation of crop yield simulated using different schemes in the United States. (a)
Maize yield in irrigated maize-growing regions of the United States, as reported by the USDA
(orange boxes), compared with simulations by CoLM using the noirrig (green boxes), irrig-unlim
(blue boxes), and irrig-lim (purple boxes) schemes. Since reported yields are at the county scale,
grid-based simulation results were aggregated to corresponding counties. (b-c) Same as (a) but
for soybean and wheat yields.
Using county-scale crop yield data for irrigated and rainfed regions provided by the USDA, we
assess simulated yields under both irrigated and non-irrigated scenarios. The dataset may not
comprehensively cover all irrigated areas in the U.S. or all years during the study period, so
comparisons are limited to regions and years with reported data. In rainfed regions, the model
broadly reproduces average annual yields for the maize, soybean, and wheat (Figure S8).
However, in irrigated regions, the model without irrigation significantly underestimates crop
yields, with average underestimations of 31.95%, 44.45%, and 35.95% for maize, soybean, and
wheat, respectively (Figure 9). Under both the irrig-umlim and irrig-lim schemes, despite slight
differences in performance across crops, the model effectively simulates yield increases under
irrigation, aligning well with observations. Differences between the two irrigation schemes are
minimal: the irrig-unlim scheme performs slightly better for maize and soybean in terms of
average biases, while the irrig-lim scheme shows better performance for wheat.
Furthermore, based on limited annual yield data, we observe that considering irrigation generally
improves the model's ability to capture inter-annual yield fluctuations (Figure S9). The KGE of



annual yields under the noirrig scheme are -1.342, -1.451, and -1.308 for maize, soybean, and
wheat, respectively, while with the irrig-umlim and irrig-lim schemes, the KGE values increase
to 0.101, -1.132, and 0.197, and -0.158, -1.449, and -0.144, respectively.

## 4. Discussions

### 4.1 Applications of the developed module

#### 4.1.1 Impacts of irrigation on energy budget

Numerous studies have highlighted the impacts of irrigation on global and regional energy
budgets and near-surface climates. In this study, we similarly examine the effects of irrigation on
the energy budget over irrigated areas in the U.S. by comparing results from the irrig-lim and
noirrig scheme. Consistent with prior research, we find that irrigation increases latent heat ($LH$)
by 7.53 W m$^{-2}$ (23.25 %) and decreases sensible heat ($SH$) by 5.18 W m$^{-2}$ (9.48 %) averaged
from 2001 to 2016, resulting in an approximately 0.30°C reduction in land surface temperature
(Figure 10). Since land-atmosphere coupling is not included, the primary mechanisms driving
these impacts are increased soil evaporation due to enhanced soil moisture and greater vegetation
transpiration driven by improved crop growth following irrigation (Figure S10 a-b). Annually,
these mechanisms contribute roughly equally to the increase in total evapotranspiration in
irrigated regions, with pronounced seasonal differences: during the peak growing seasons
(summer and autumn), the contribution was dominated by vegetation transpiration, while in other
seasons, particularly winter, the increase in soil evaporation plays a larger role in affecting
regional energy distribution and temperature (Figure S10c).
This study further explores the spatial characteristics of these impacts, analyzing the correlations
between irrigated area, irrigation water withdrawal, and changes in $LH$, $SH$, and land surface
temperature ($\Delta LH$, $\Delta SH$, $\Delta T_s$) across different climate zones. Notably, irrigation has the most
substantial impact in arid regions, especially on $LH$, where $\Delta LH$ is more than double that of
semi-arid and humid regions, with a larger reduction in temperature by 0.36°C. Interestingly,
while previous studies have emphasized irrigated area as the primary determinant of irrigation-
induced climate effects (Al-Yaari et al., 2022; Chen and Dirmeyer, 2019), our results indicate
that irrigation water withdrawal has a stronger influence on the regional energy budget and
temperature. Across all climate zones, $\Delta SH$, $\Delta LH$, and $\Delta T_s$ are significantly correlated ($p < 0.01$)
with irrigation water withdrawal, with correlation coefficients of -0.81, 0.79, and -0.82,
respectively (Figure 10 (b, e and h)), which are higher than the correlations with irrigated area (-
0.59, 0.61, and -0.52; Figure 10 (c, f and i)). This emphasizes the critical role of water
availability in modulating the climate effects of irrigation.

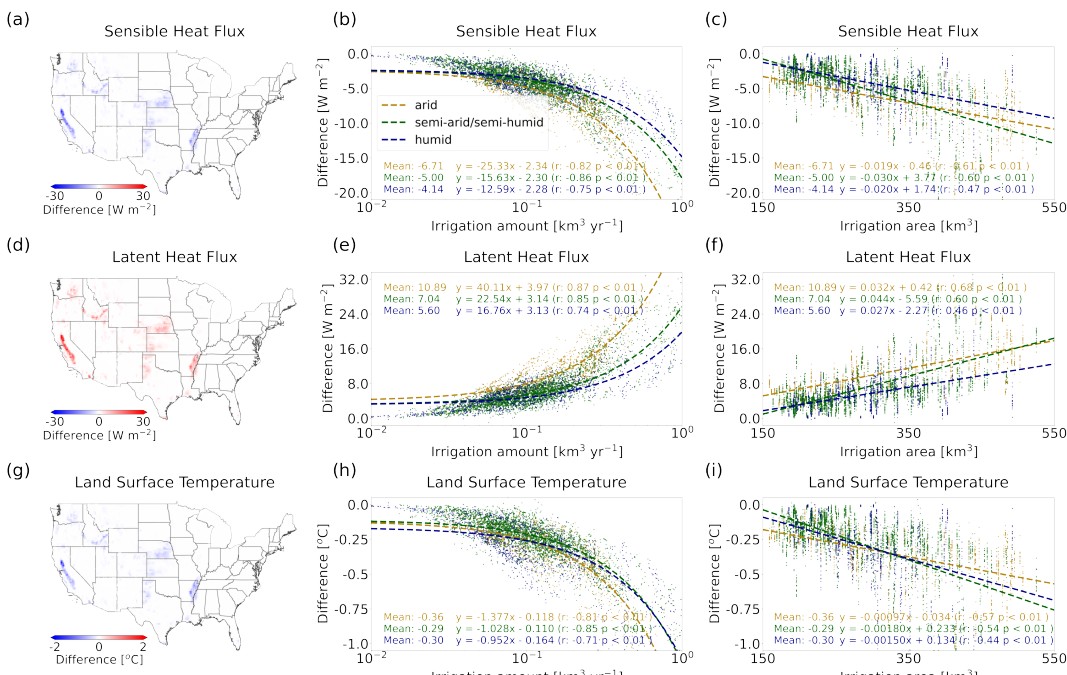


**Figure 10.** Impact of irrigation on local energy flux and surface temperature in the United States.
(a) Impact of irrigation on sensible heat flux, quantified by the difference ($\Delta SH$) between the
noirrig and irrig-lim simulation results. (b) Relationship between irrigation amount and $\Delta SH$,
with grid colors indicating the climate zones (i.e., arid, semi-arid/semi-humid, humid). For each
climate zone, the mean $\Delta SH$, the regression line of irrigation amount versus $\Delta SH$, and the
regression equation are displayed. (c) Same as (b), but for the relationship between irrigation
area and $\Delta SH$. (d-f) Same as (a-c), but for the impact on latent heat flux ($\Delta LH$). (g-i) Same as (a-
c), but for the impact on land surface temperature ($\Delta T_s$).

It is important to note that this study employs offline land simulations and does not account for
land-atmosphere interactions, which may introduce biases in the estimated climate impacts.
Future studies should include coupled land-atmosphere simulations to provide a more
comprehensive assessment (Cook et al., 2015; Puma and Cook, 2010; Sacks et al., 2009).
Another aspect worth considering is that some farmers irrigate not only to address water deficits
but also to mitigate heat stress during high-temperature periods (Verma et al., 2020). This
practice can notably affect local temperatures. For instance, surface water temperatures generally
track air temperatures, whereas groundwater temperatures remain relatively stable throughout the
year—typically warmer than air in winter and cooler in summer. This temperature difference,
especially in regions relying on groundwater irrigation, may have non-negligible effects on local
climate that should be incorporated into future modeling efforts.



### 4.1.2 Assessments of irrigation water security


This study compares the irrigation schemes with and without water availability constraints,
highlighting the necessity and importance of incorporating water limitations into simulations.
Our results demonstrate that including these constraints improves simulation accuracy,
particularly in the modeling of water systems. Specifically, irrigation water withdrawal simulated
under the irrig-lim scheme aligns more closely with observational data (Figure 3 and Figure 6).
Validation against river flow observations further supports the improved performance of the
irrig-lim scheme. Importantly, this scheme avoids the risk of potential water imbalances in the
modeled hydrological system—an issue commonly associated with non-constrained schemes
(Figure 8).
Additionally, incorporating water availability constraints more accurately reflects the reality of
water resource utilization. By accounting for the interconnections between subsystems within the
irrigation water demand-supply system, this approach enables simulation and prediction of
irrigation water security issues. Here, we visualize the average number of days when water
supply was insufficient to fully meet irrigation demand that simulated by the irrig-lim scheme
(Figure 11). Spatially, in humid regions, where irrigation demand is low and water resources are
abundant, fewer days of unmet irrigation needs occur. Conversely, in arid regions, where
irrigation demand is high and water resources are often limited, the number of unmet irrigation
days increases significantly. Figure 11a illustrates that states with a higher number of unmet
irrigation days are also those with relatively scarce water resources (e.g., Montana and Nevada).
From a temporal perspective, drought years lead to increased irrigation requirements due to
reduced precipitation or higher evaporative demand. Although additional water withdrawals can
partially address this increased demand, drought conditions often simultaneously result in
deficits in both surface and groundwater resources within the water system. As a result, most
states experience a substantial increase in unmet irrigation days during drought years (an average
of 43 days). In contrast, during wetter years, the number of unmet days decreases significantly
(an average of 31 days).
Reported disaster data show that even with irrigation, significant crop losses can occur during
drought years, aligning with broader water security challenges (Mieno et al., 2024). Our
approach effectively captures this phenomenon by describing the connectivity between
subsystems in the water demand-supply system and highlighting the impact of water limitations
on irrigation. In contrast, ignoring these constraints risks underestimating potential food security
issues in a future characterized by more frequent and/or severe droughts. This represents a
critical limitation of crop and land surface models that adopt irrigation schemes without
considering water availability constraints.





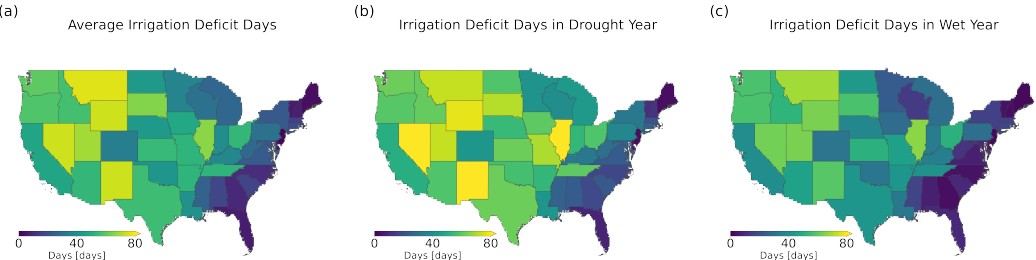


**Figure 11.** Days per year with unmet irrigation demand (i.e., irrigation deficit days) in the United
States simulated by the irrig-lim scheme. (a) Multi-year average irrigation deficit days from 2001
to 2015 for individual states. (b) Irrigation deficit days in drought year for individual states. (c)
Irrigation deficit days in wet year for individual states. Drought year (wet year) is defined as the
year with the lowest (highest) annual precipitation during 2001–2016.

**4.2 Limitations and a way forward**

While the developed module represents a significant advancement in modeling irrigation water
system within land surface models by providing a comprehensive representation of the irrigation
processes—including water demand, water withdrawal, and water utilization, several limitations
and assumptions should be acknowledged.

Irrigation water demand in this study is estimated using a soil moisture deficit method. However,
the parameterization of certain key variables (e.g., target and threshold soil moisture levels) is
overly simplified and does not account for variations among crop types. These parameters are
adjustable, and their calibration could further enhance the model's accuracy in reproducing
irrigation water use. Additionally, in some cases, farmers irrigate not only to address soil
moisture deficits but also to reduce crop heat stress during high-temperature periods—a factor
that should be incorporated into future modeling efforts. Furthermore, this study did not account
for water losses during conveyance and application. Irrigation losses, as noted by Jägermeyr et
al. (2015), include conveyance losses and on-field application losses. By ignoring conveyance
losses, the model assumes that water withdrawn equals water applied, likely leading to an
underestimation of total irrigation water use. Field application losses depend on irrigation
methods (Leng et al., 2017), and while this study considered four irrigation systems with
differentiated efficiencies, the reliance on simplified rules and a coarse irrigation map fails to
reflect the diversity of irrigation methods and distributions. For example, actual sprinkler
systems distribute water in specific spray patterns rather than uniformly. However, the model
assumes uniform water distribution across each Crop Functional Type (CFT). Future models
could benefit from parameterizations that capture spatial heterogeneity in irrigation distribution
(Jägermeyr et al., 2015; Merriam et al., 1999). Moreover, irrigation water demand also depends
on agricultural practices, such as crop types, cropping calendars, and planting intensities. While
the model determines crop phenology based on meteorological data, real cropping calendars are



influenced by farmers' decisions (Sacks et al., 2010). Incorporating satellite-derived phenology
data could better represent these human factors. Addressing these agricultural practices is crucial
for improving the accuracy and applicability of irrigation models.
In simulations of irrigation water withdrawal, this study provides a detailed representation of
reservoir water withdrawal but acknowledges several sources of uncertainty: First, the dataset
includes fewer dams than exist, as it focuses primarily on large dams and may lack data due to
protection policies. This omission likely contributes to the underestimation of surface water
extraction in some states. Second, all dams are assumed to supply irrigation water, although
some reservoirs may not serve this purpose. The irrigation areas served by each dam are
unknown, and a generalized estimation method is employed in this study, introducing large
uncertainties that remain difficult to validate. Third, dam operations are simplified, while in
reality, they often involve complex considerations, such as multi-objective operations and
coordinated management of multiple reservoirs. Advanced reservoir optimization strategies,
which require predictive simulations and prior knowledge of future inflows and demands, are not
incorporated into the model, presenting a significant challenge for considering the impacts of
complex human decision-making in land surface models.
Determining the division of irrigation water withdrawals between surface and groundwater
sources, as well as the withdrawal sequence, is also critical. This study allocates irrigation
demand based on pre-defined proportions and simultaneously withdraws water from both
sources. Surface water demand is met sequentially through local runoff, river discharge, and
upstream reservoir storages. This method, employed in models such as ORCHIDEE v2.2
(Arboleda-Obando et al., 2024) and E3SM (Zhou et al., 2020), provides satisfactory simulations
of water source allocation for irrigation (Figure 4 vs. Figure S11). However, its reliability
depends on the accuracy of input data and may underestimate withdrawals if any water source is
inadequately represented. Alternatively, some models (e.g., MATSIRO and CLM5; Pokhrel et al.,
2012; Yao et al., 2022) do not pre-allocate demand but set a fixed order of water withdrawals—
typically prioritizing surface water before groundwater. This method tends to satisfy more
irrigation demand and provides better estimates in regions with unreported groundwater
extraction. We propose that a hybrid approach, defining surface and groundwater proportions
dynamically, warrants consideration in future study. For instance, during wet seasons, surface
water extraction proportions could increase to reduce groundwater reliance and associated
pumping costs. Conversely, during dry seasons, surface water may be more constrained,
necessitating greater reliance on groundwater for irrigation. However, such an approach still
needs to address challenges, including unreported groundwater use, data scarcity, and the
physical, technical, and socio-economic constraints on groundwater use across regions.
Additionally, this study does not account for restrictions beyond water availability, such as local
regulations governing water allocation, including water rights and inter-basin water transfers.
Alternative water sources, such as desalinated seawater and treated wastewater, also warrant
consideration (Vliet et al., 2021). Recent assessments indicate that these unconventional water





sources are experiencing exponential growth (Jones et al., 2019). Although their contributions

remain low globally, they play a significant role in water-scarce regions. Incorporating these
factors into models could further improve simulations of irrigation water security.
Finally, the developed module's results and applicability are strongly influenced by the CoLM
framework itself. A critical aspect requiring careful consideration is the evaluation and
calibration of hydrological variables, such as soil moisture, runoff, river discharge, and
groundwater levels, which are essential for water resource modeling. Currently, the CoLM
employs the simplified top model (SIMTOP) developed by Niu et al. (2005) for runoff
simulations. The excessive simplification of this approach, coupled with the lack of calibration,
limits the model's accuracy in runoff simulations. Inadequate representation of snow and glacial
melt processes introduces regional biases, particularly in northern and midwestern U.S. states
where these factors are pivotal. For instance, surface water extraction is underestimated in some
states within these regions, likely because the model fails to accurately capture snowmelt and
glacial melt contributions to streamflow, leading to erroneous estimates of surface water
availability. Similarly, simulated evapotranspiration is systematically underestimated, even in
areas without crops or irrigation, likely due to more complex underlying causes. These biases,
when aggregated at the watershed level, result in significant discrepancies in river discharge,
thereby constraining the model's applicability for water resource management and its ability to
predict irrigation water security. Addressing these issues requires urgent improvements in the
representation of related processes, along with further calibration and parameter tuning.
**5. Conclusions**
The growing challenges posed by increasing global food demand and water scarcity underscore
the need for advanced modeling tools capable of accurately capturing human-water interactions.
This study contributes to addressing this need by implementing a two-way coupled irrigation
water system within the Common Land Model. The developed module provides a
comprehensive representation of the entire irrigation water use process, including water demand,
withdrawal, and utilization. It introduces a refined multi-source water withdrawal framework and
achieves bidirectional coupling between water demand and withdrawal during simulation.
The robustness of the new irrigation module is validated through simulations across the
contiguous United States, focusing on regional-scale water, energy, and crop yield dynamics. The
module effectively simulates total national annual irrigation withdrawals, their spatial
distribution, the proportions of different water sources, and irrigation volumes for various crops.
Compared to other hydrological models in ISIMIP2a, our model performs similarly or better in
simulating U.S. irrigation withdrawals. Incorporating the new irrigation module also
significantly improves the accuracy of simulated surface energy fluxes, both in magnitude and
seasonal patterns, resulting in more accurate surface temperature predictions. For streamflow, the
irrigation scheme accounting for water availability constraints enhances the model's
representation of hydrological system dynamics, whereas the unrestricted irrigation scheme



introduces potential water budget imbalances. Additionally, the new module markedly improves
the model's ability to simulate annual yields and interannual fluctuations of major crops,
including maize, soybean, and wheat.
We further apply the developed module in two novel analyses. First, the scheme effectively
characterizes the climatic impacts of irrigation, revealing a stronger positive correlation between
irrigation water volume, rather than irrigated area, and the intensity of irrigation-induced climatic
effects. This highlights the critical role of water availability in modulating irrigation-driven
climate impacts. Although more accurate simulation of these effects requires land-atmosphere
coupled modeling, the enhanced CoLM is clearly ready for such tasks. Second, the module
captures irrigation-related water security issues, particularly during drought years, where water
shortages across the resource system lead to irrigation water deficits and associated food security
challenges. These results demonstrate the promise of CoLM as a valuable tool for future water
use and scarcity assessments, paralleling the functionality of global hydrological models and
contributing to initiatives such as the Inter-Sectoral Impact Model Intercomparison Project.
**Data Availability Statement**
The meteorological variables from the WFDEI can be freely accessed from
ftp://rfdata:forceDATA@ftp.iiasa.ac.at. The land cover type datasets (MCD12Q1) can be freely
accessed from https://lpdaac.usgs.gov/products/mcd12q1v061/. The soil characteristics datasets
(GSDE) can be freely accessed from http://globalchange.bnu.edu.cn/research/data/. The
CropScape and Cropland Data Layer (CDL) datasets can be freely accessed from
https://nassgeodata.gmu.edu/CropScape/. The crop calendar datasets can be freely accessed from
https://zenodo.org/records/5062513/. The irrigation map and irrigation equipment percentage can
be freely accessed from https://www.fao.org/aquastat/en/geospatial-information/global-maps-
irrigated-areas/latest-version/. The GRanD database can be freely accessed from
https://www.globaldamwatch.org/grand/. The GRSAD database can be freely accessed from
https://dataverse.tdl.org/dataset.xhtml?persistentId=doi:10.18738/T8/DF80WG/. The ReGeom
database can be freely accessed from https://zenodo.org/records/1322884/. The USGS's
hydrological survey data can be freely accessed from https://water.usgs.gov/watuse/data/. The
USDA NASS's agricultural survey data can be freely accessed from
https://quickstats.nass.usda.gov/. The crop-specific irrigation water withdrawals data can be
freely accessed from https://doi.org/10.13012/B2IDB-2656127_V1/. The ISIMIP2a datasets can
be freely accessed from https://data.isimip.org/search/. The FluxCom datasets can be freely
accessed via ftp.bgc-jena.mpg.de. The ERA5-Land skin temperature data can be freely accessed
from https://cds.climate.copernicus.eu/datasets/reanalysis-era5-land-monthly-
means?tab=download/. The streamflow data (GRDC) can be freely accessed from
https://www.bafg.de/GRDC/EN/Home/. CoLM codes are available for download from GitHub
(https://github.com/CoLM-SYSU/CoLM202X/).



**Author contributions**

SZ and YD conceptualized and designed the study. SZ and HL collected the data, developed the model, and conducted the analyses. FL and XL provided assistance with model development. HL prepared the figures. SZ drafted the manuscript. All authors contributed to result interpretation and reviewed the final manuscript.

**Competing interests**

The authors declare no competing interests.

**Acknowledgments**

This research was supported by the Guangdong Major Project of Basic and Applied Basic Research (Grant No. 2021B0301030007), the National Natural Science Foundation of China (Grant Nos. 42088101 & 42175168), the Guangdong Natural Science Foundation (Grant No. 2023A1515011541), the Innovation Group Project of Southern Marine Science and Engineering Guangdong Laboratory (Zhuhai) (Grant No. 311022006) and the Fundamental Research Funds for the Central Universities, Sun Yat-sen University (Grant No. 241gqb007). We also acknowledge the high-performance computing support from the School of Atmospheric Science of Sun Yat-sen University.

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
