# Peer review of "Representation of a two-way coupled irrigation system in the Common Land Model"

_EGUsphere, 2024_

## Referee Comment (RC2)

This study contributes to the field of irrigation modeling by incorporating an irrigation scheme into the Common Land Model (CoLM) by building upon established methodologies from existing literature. I believe this article has the potential for publication in *Hydrology and Earth System Sciences* (HESS). However, there is still much room for improvement in model validation. My comments are outlined below.

Major Comments

1. Before introducing the "Two-way coupled irrigation water use module," it would be beneficial to provide a brief overview of the Common Land Model (CoLM)'s water and energy processes related to irrigation. This will help readers better understand CoLM's key mechanisms and how the new module integrates with them.

2. The irrigation system employed in this study is based on the soil moisture deficit method. The authors need to explain how this strategy is appropriate for the study region. Specifically, are the parameters $f_{irrig}$ and $f_{thresh}$ set to their default values? If so, using field capacity or saturation water amount as the target and threshold values for the root zone (1m) may be impractical, as it would theoretically demand an excessively large volume of water (as illustrated in Figure 3). While subsequent limitations might arise due to water availability constraints, this process remains inherently passive, as it is predicated on an initially overestimated assumption. If not, I suggest listing the value of the

parameters.

3. Model evaluation. Terrestrial Water Storage (TWS) anomaly is a crucial variable for model evaluation. Comparing this variable with GRACE satellite data would enhance the study's robustness and provide additional validation for the model's performance.

Minor Comments

1. Experimental Design. Consider creating a table that clearly delineates the differences between each experiment. Specifically, the table should highlight the scheme-specific variations in surface water and groundwater supply between the two irrigation experiments. This would help improve clarity and facilitate comparisons.

2. Figure 7. Although the simulations of latent heat flux and sensible heat flux have improved, there is still a discrepancy compared to FLUXCOM data. Additionally, the fact that irrigation leads to an underestimation of temperature from June to September needs further discussion.

3. References. Some citations share the same author surname and publication year, which may cause linking issues. To resolve this, distinct labels (e.g.,2024a, 2024b) should be added, or additional author names can be included to differentiate them.

4. Figure S1. (1) Please confirm that the units in the figure ($km^3$ -> $km^2$ ?); (2) The color bar range needs to be adjusted. Due to at mid-latitudes

(around 40°N), a 0.25°×0.25° grid cell covers approximately 770 km². Alternatively, you could use percentages to present the data for better clarity.

5.  Since the validation of simulated irrigation is obtained from the USGS. I suggest including some supporting text related to USGS and show total irrigation water withdrawals and categorized by surface and groundwater sources in Table format as a supplementary text.

---

## Author Comment (AC1)

**Response to Reviewers' Comments**

We greatly appreciate the editor and the reviewer for providing valuable and constructive comments on our manuscript (egusphere-2024-4093). We seriously considered each comment and revised the manuscript accordingly. The individual comments are addressed in the following response letter and the manuscript has been revised to accommodate the changes. Below are our detailed responses, with the comments from the editor and the reviewer in black followed by our responses in blue. Please note that the comments are marked with codes for brevity, such as R1C2 (Comment 2 from Reviewer 1).

R1C1: In the manuscript entitled "Representation of a two-way coupled irrigation system in the Common Land Model", the authors implemented several new features regarding irrigation water applications and withdrawals in a land system model and evaluated the model with new features based on observation-based datasets. Although these new features are mainly adopted from other models, it is impressive that the authors implemented them in a different land model. I congratulate the authors for this huge amount of work and believe that it is qualified to be published on EGU HESS. However, there is still some room for improvement in the manuscript, which are listed below.

Response: We sincerely appreciate your thoughtful review and encouraging feedback. Your insights have been invaluable in improving the manuscript. Below, we provide detailed point-by-point responses to your comments, along with the corresponding revisions in the manuscript.

R1C2: Before introducing the crop module, it would be better to first describe the sub-grid cell level structure of the model CoLM. How are the calculations conducted over different land-use tiles? How are rainfed and irrigated cropland differently treated? Etc.

Response: Thanks for your valuable suggestion. We have added a detailed description of the sub-grid cell level structure in CoLM, including how calculations are performed across different land-use tiles and the distinction between rainfed and irrigated cropland. Please see Line 173-199 in the revised manuscript and Figure S1 in the revised supplement.

Lines 179-199: *"In CoLM2024, the 'patch' serves as the fundamental computational unit to account for land surface heterogeneity (Figure S1). Based on land type, patches are divided into five types: vegetation (including bare soil), urban, wetlands, glaciers, and water bodies. The vegetation patch is further classified into natural*

*vegetation and crops, represented using the Plant Functional Type (PFT) approach. Under this framework, all natural vegetation within a grid cell is treated as a single patch, sharing common soil thermal and moisture conditions while radiative and photosynthesis processes are simulated independently. When the crop model is activated, each crop type (distinguishing between rainfed and irrigated crops) is treated as an independent patch. This means that the calculations of soil moisture and thermal processes for each crop patch remain independent, without shared water and heat dynamics.*

*At each patch, the primary thermal processes include precipitation phase change, radiation transfer, temperature calculations for leaves, snow, and soil, turbulent exchange, etc. The key hydrological processes include canopy interception, evapotranspiration, surface runoff, infiltration, soil water vertical movement, subsurface runoff, groundwater, river routing, etc. Specifically, the two-big-leaf scheme is employed to compute radiation transfer, leaf temperature, photosynthesis and transpiration (Dai et al., 2004; Yuan et al., 2017). Surface turbulent exchange is simulated using similarity theory (Liu et al., 2022; Zeng and Dickinson, 1998). Soil and snow temperature are determined using the heat diffusion equation, considering only vertical exchange (Dai and Yuan, 2014). Canopy interception is calculated same as CoLM2014 with considering the leaf angle and precipitation phase (Dai and Yuan, 2014; Sellers et al., 1996). Soil water vertical movement is simulated by the Richards equation and Buckingham-Darcy's law with using the Campbell soil water characteristic curve scheme to close the Richards equation (Buckingham, 1907; Campbell, 1974; Richards, 1931). Surface and subsurface runoff are estimated using the SIMTOP approach (Niu et al., 2005). When the irrigation scheme is activated, irrigation water is applied to the canopy or top soil according to predefined irrigation methods and simulated irrigation amounts, thereby influencing the soil moisture and thermal processes within the irrigated patches."*

[Figure]

**Figure S1.** Diagram of the sub-grid structure in the Common Land Model.

R1C3: The authors adopted the crop model used in the Community Land Model version 5 (CLM5) and activated the crop model, which means that the crop phenology is simulated. I think it is important to also compare the simulated and observed LAI at some single points to show how the model reproduces the phenology of different crop types.

Response: As per your suggestion, we have added a validation of simulated LAI using observed LAI from flux tower sites within the study area (see Lines 123-132 in the revised supplement).

Line 123-132: *"We selected multiple crop sites from FLUXNET and AmeriFlux, with details provided in the Table S4, including only stations where the same crop had been sown for more than two years. The results indicate that the model effectively captures the seasonal dynamics of LAI across different sites, regardless of whether the crops are rainfed or irrigated (Figures S20 and S21). However, LAI values were underestimated at certain site years, such as US-Ne3 in 2002 and 2006, when rainfed soybean was planted (Figure S20 (d and f)). The underestimation is primarily due to the proximity of US-Ne3 to irrigated sites (US-Ne1 and US-Ne2), where soil moisture conditions may be influenced by nearby irrigation. In contrast, the simulated LAI for rainfed soybean at US-IB1 closely aligns with observed values."*

Table S4. Stations information.

| station | location | LAI years | crop type | irrigation management |
|---|---|---|---|---|
| US-Ne1 (Suyker, 2024a) | 41.18N, 96.44W | 2002; 2004; 2006 | maize | irrigated |
| US-Ne2 (Suyker, 2024b) | 41.16N, 96.47W | 2002, 2004, 2006 | soybean | irrigated |
| US-Ne3 (Suyker, 2024c) | 41.18N, 96.44W | 2001, 2003, 2005 | maize | rainfed |
| US-Ne3 (Suyker, 2024c) | 41.18N, 96.44W | 2002, 2004, 2006 | soybean | rainfed |
| US-IB1 (Matamala, 2019) | 41.86N, 88.22W | 2005; 2007 | soybean | rainfed |
| US-ARM (Biraud et al., 2024) | 36.61N, 97.49W | 2005; 2008 | maize | rainfed |
| US-ARM (Biraud et al., 2024) | 36.61N, 97.49W | 2002; 2008 | winter wheat | rainfed |

[Figure]

**Figure S20.** Comparison of reported and simulated LAI phenology at rainfed stations. (a) US-Ne3 for maize in 2001, as reported by the AmeriFlux (red dots), compared with simulations by CoLM without irrigation (green line). (b-c) Same as (a) but in 2003 and 2005. (d-f) Same as (a) but for soybean in 2002, 2004 and 2006. (g) and (j) Same as (a) but for maize at US-ARM in 2005 and 2008. (h) and (k) Same as (a) but for soybean at US-IB1 in 2005 and 2007. (i) and (l) Same as (a) but for winter wheat at US-ARM in 2002 and 2008.

[Figure]

**Figure S21.** Comparison of observed and simulated LAI phenology at irrigated stations. (a-c) US-Ne1 for maize in 2002, 2004 and 2006, as reported by the AmeriFlux (red dots), compared with simulations by CoLM with irrigation (green line). (d-f) Same as (a-c) but for soybean at US-Ne2 in 2002, 2004 and 2006.

R1C4: I do not think comparing the simulated and observed energy fluxes averaged over all irrigated regions in the USA is enough. I would compare them over single grid cells with intense irrigation, or at least, average across the irrigated regions in different climate zones.

Response: In the manuscript, the original Figure 7b, d, f already present kernel density plots of the KGE between observed and simulated fluxes at each irrigation grid cell. As per your suggestion, we have further included spatial distribution maps of bias and KGE between observed and simulated energy fluxes at each irrigated grid cell. Please refer to Figures S8 and S9 in the revised supplement. Additionally, we have incorporated a corresponding description in the main text:

Lines 583–585: *"With the inclusion of the irrigation module, simulation errors in surface energy fluxes over irrigated areas are significantly reduced, particularly in the U.S. High Plains and the California Central Valley (Figures S8 and S9)."*

[Figure]

**Figure S8.** Evaluation of simulated energy fluxes and land surface temperature in the irrigation region. (a) Bias between observed monthly sensible heat flux and simulations from CoLM under the noirrig scheme in irrigation regions of the United States. (b) Same as (a) but for irrig-unlim scheme. (c) Same as (a) but for irrg-lim scheme. (d-f) Same as (a-c) but for latent heat flux. (g-i) Same as (a-c) but for or land surface temperature.

[Figure]

**Figure S9.** Evaluation of simulated energy fluxes and land surface temperature in the irrigation region. (a) The Kling-Gupta efficiency (KGE) between observed monthly sensible heat flux and simulations from CoLM under the noirrig scheme in irrigation regions of the United States. (b) Same as (a) but for irrig-unlim scheme. (c) Same as (a) but for irrg-lim scheme. (d-f) Same as (a-c) but for latent heat flux. (g-i) Same as (a-c) but for or land surface temperature.

R1C5: Similarly, I would do some evaluations of streamflow over some bigger catchments (which can consist of several sub-catchments that the authors used) with intense water withdrawal. In the supplementary Figure S7, it would be nice to also plot the boundaries of catchments. A map showing the difference between simulated and observed streamflow will be better than the bar plots in Figure 8.

Response: Thank you for your valuable suggestions. Following your suggestions, we have added catchment boundaries to Figure S10, included spatial distribution maps to illustrate the differences between simulated and observed streamflow (Figure S11), and evaluated streamflow over larger catchments with intense water withdrawals to provide a more intuitive assessment of model performance (Figure S12 and Lines 641–645).

Lines 641–645: *"Furthermore, we compared observed and simulated monthly streamflow in ten larger catchments (Figure S12), providing a more intuitive*

*assessment. The results clearly indicate that the irrig-lim scheme produces streamflow estimates that align more closely with observations, whereas the irrig-unlim scheme tends to overestimate streamflow, particularly during months with high irrigation demand."*

[Figure]

**Figure S10.** Locations of catchment outlets and boundaries of the 77 irrigation-affected catchments.

[Figure]

**Figure S11.** Evaluation of simulated streamflow in 77 irrigation-affected catchments. (a) Percentage bias (PBIAS) between observed monthly streamflow and simulations from CoLM under the noirrig scheme for each catchment. (b) Same as (a) but for irrig-unlim scheme. (c) Same as (a) but for irrg-lim scheme. (d-f) Same as (a-c) but for the Kling-Gupta efficiency (KGE) between simulated and observed streamflow.

[Figure]

**Figure S12.** Evaluation of simulated streamflow in 10 large irrigation-affected catchments. (a-j) Monthly streamflow averaged from 2001 to 2016 for each catchment, based on GRDC dataset (red lines) and simulated by CoLM using the noirrig (green lines), irrig-unlim (blue lines), and irrig-lim schemes (purple lines). (k) Boundaries of the selected 10 irrigation-affected catchments (red lines).

R1C6: For the crop yield, I would also like to see a map showing the difference between simulated and observed crop yield in administrative regions, rather than the bar plot in Figure 9.

Response: As per your suggestion, we have added spatial distribution maps illustrating the differences between observed and simulated crop yields at the county scale (Figure S15). We have also incorporated a corresponding description in the main text:

Line 703-707: "*As shown in Figure S15, the yield underestimation observed in most counties without irrigation is substantially corrected under the irrig-unlim (and irrig-lim) schemes, with 90.5% (70.8%), 99.5% (94.2%), and 68.4% (74.8%) of counties showing absolute yield differences for maize, soybean, and wheat within 1 t ha⁻¹ compared to observations.*"

[Figure]

**Figure S15.** Evaluation of simulated crop yield in the irrigation region. (a) Bias between observed maize yield and simulations from CoLM under the noirrig scheme in irrigation regions of the United States. (b) Same as (a) but for irrig-unlim scheme. (c) Same as (a) but for irrg-lim scheme. (d-f) Same as (a-c) but for soybean yield. (g-i) Same as (a-c) but for or wheat yield.

R1C7: It is always recommended to add maps showing difference between simulations and observations in Figure 3 and 4.

Response: As per your suggestion, we have added spatial distribution maps illustrating the differences between simulations and observations. Please see Figure S8, S9, S11, S12, S13 and S15 in the revised supplement.

R1C8: It would be interesting to add some results showing the new features of CoLM irrigation module. For example, showing the simulated water withdrawal from different sources, showing simulated terrestrial water storage, showing how lakes or reservoirs change with irrigation water withdrawal (if they do), etc. This will help readers to know what can they use the model to do.

Response: We appreciate your constructive comments. To illustrate the potential applications enabled by the developed irrigation module in CoLM, we had included two examples in the discussion section. In Section 4.1.1 *Impacts of irrigation on the*

*energy budget*, we analyzed the effects of irrigation on the regional energy budget in the U.S. by comparing results from the irrig-lim and noirrig schemes. Our findings indicate that irrigation water withdrawal exerts a stronger influence on the energy budget and temperature than irrigated area, underscoring the critical role of water availability in modulating irrigation-induced climate effects (Figure 11). In Section 4.1.2 *Assessments of irrigation water security*, we demonstrated that the irrig-lim scheme effectively captures an increase in the number of days per year with unmet irrigation demand during drought years due to overall water scarcity (Figure 12), highlighting its potential for evaluating irrigation water security.

We understand that you may be particularly interested in the module's capability to capture spatiotemporal variations in water resources. Regarding water withdrawal from different sources, Figure 4 already demonstrates the model's ability to simulate surface and groundwater withdrawals. To further illustrate the distribution of surface water withdrawals, we have added Figure S18, which shows the fraction of withdrawals sourced from local water resources (runoff and river streamflow) and upstream reservoirs. However, this result currently lacks observational data for validation. Regarding terrestrial water storage, we have incorporated an evaluation of the model's performance in simulating terrestrial water storage anomalies by comparing the results with GRACE satellite data. A description of the GRACE dataset has been added (Lines 439-442), and the validation results are presented in the revised manuscript (Lines 656-683, Figure 9, and Figure S13).

[Figure]

**Figure S18**. Comparison of reported and simulated annual irrigation water withdrawal by water source. (a) Annual withdrawal amounts from different sources for the top 20 states by irrigation water withdrawal, using data from USGS reports. (b) Same as (a), but for simulated by CoLM using the irrig-lim scheme.

Lines 439-442: *"For terrestrial water storage (TWS) validation, we utilized monthly terrestrial water storage anomaly data from the Gravity Recovery and Climate Experiment (GRACE) mission for the period 2002-2016, with a spatial resolution of 0.5 degree, provided by the NASA Jet Propulsion laboratory (Watkins et al., 2015; Wiese et al., 2016)."*

Lines 656-683:

*"**3.2.3 Evaluation of simulated terrestrial water storage anomalies**

To assess the model's ability to simulate the impact of irrigation on terrestrial water storage (TWS) dynamics, we compared the simulated monthly TWS anomalies with*

*those derived from GRACE satellite products provided by the NASA Jet Propulsion laboratory. The results showed that incorporating the irrigation module, particularly under the irrig-lim scheme, improved the model's ability to capture both the interannual variability (Figure 9a) and seasonal patterns (Figure 9b) of TWS anomalies. Under the noirrig scheme, the simulated monthly TWS anomalies from 2002 to 2016 had a Pearson correlation of 0.25 with GRACE data and an RMSE of 6.75. In contrast, the irrig-lim scheme increased the correlation to 0.75 and reduced the RMSE to 5.13 (Figure 9a). The spatial distribution of Pearson correlation coefficients between the simulations and GRACE data (Figure S13) further demonstrated a widespread improvement across the U.S., particularly in the Corn Belt.*

*The enhancement was even more pronounced in the simulation of seasonal TWS anomaly patterns. Without irrigation, the model underestimated seasonal variations, resulting in a pattern that deviated substantially from GRACE observations. This bias was effectively corrected in the irrig-lim scheme, where the Pearson correlation coefficient increased to 0.92 and the RMSE decreased to 3.44 (Figure 9b). However, none of the simulations captured the decline in GRACE-derived TWS anomalies during 2012 to 2016, likely due to groundwater depletion from irrigation (Rodell and Reager, 2023). This suggests that the model may require further validation and improvements in simulating irrigation-induced groundwater storage changes."*

[Figure]

**Figure 9.** Evaluation of simulated terrestrial water storage anomalies in irrigated region. (a) Time series of monthly terrestrial water storage anomalies from 2001 to 2016, simulated by CoLM (under the noirrig, irrig-unlim, and irrig-lim schemes) and derived from GRACE (JPL dataset). The Pearson correlation coefficient (r) and root mean square error (RMSE) between the simulations and GRACE data are indicated in the panel. (b) Climatological monthly terrestrial water storage anomalies averaged over 2001–2016, simulated by CoLM and derived from GRACE.

[Figure]

**Figure S13.** Comparison of observed and simulated monthly terrestrial water storage anomalies in the United States. (a) Spatial distribution of the Pearson correlation coefficient (r) between GRACE-derived TWS anomalies (JPL dataset) and CoLM simulations under the noirrig scheme. (b–c) Same as (a) but for the irrig-unlim and irrig-lim schemes, respectively.

R1C9: I have to admit that I am a bit disappointed that all the evaluation work was done solely in the USA. I understand that in the USA there are generally more abundant observations, but it should at least be discussed in the discussion that some efforts are needed if other researchers want to use this model for studies over other regions.

Response: We appreciate your thoughtful comment. While our evaluation is primarily focused on the United States due to data availability, the model framework is designed to be applicable globally. To address this limitation, we have added a discussion on key considerations for applying the coupled model in other regions. Please see Lines 907–919 in the revised manuscript.

Lines 907-919: *"Although the validation in this study is limited to the United States, the framework can be applied to other regions with appropriate datasets. For example, the module requires defining the allocation and sequence of withdrawals from surface water and groundwater sources. When applying the model to other regions, the groundwater and surface water withdrawal ratios should be pre-defined based on local infrastructure, such as groundwater extraction facilities, and the withdrawal sequence can also be adjusted accordingly. For surface water withdrawals, it is essential to prepare river network data and reservoir information for CaMa-Flood simulations. Additionally, improving simulation accuracy may require incorporating region-specific crop distribution (distinguishing rainfed and irrigated areas), crop characteristics (e.g., phenology, photosynthetic capacity, carbon allocation), and management practices (e.g., planting dates, irrigation strategies). Furthermore, CoLM offers multiple parameterization schemes for*

*thermal, hydrological, and biogeochemical processes, which should be evaluated for their suitability in the target region.''*

R1C10: L193 What is this crop growth stage? When crop LAI > 0?

Response: As stated in Line 205-206, the model divides crop growth into three stages: sowing to emergence, emergence to grain filling, and grain filling to maturity. Irrigation is triggered from emergence to maturity, corresponding to the period when LAI is greater than zero.

R1C11: L205-207 Is this coefficient identical in all grid cells? Or can we assign different values in different regions?

Response: In this study, the parameters ($f_{irrig}$ and $f_{thresh}$) are set to their default value of 1, as now clarified in the revised manuscript (Line 241). However, these parameters can be adjusted for different regions based on observational data or specific research need.

R1C12: L221-222 A table showing the main difference among them would be helpful.

Response: As per your suggestion, we have added a supplementary table to summarize the key differences among various irrigation methods. Please see Table S1 in the revised supplement.

**Table S1.** Key differences among various irrigation methods.

| Feature | Drip | Sprinkler | Flood | Paddy |
|---|---|---|---|---|
| Irrigation trigger $\theta_{trigger}$ | $\Phi_{sfc}$* | $\Phi_{sfc}$ | $\Phi_{sfc}$** | $\Phi_0$ |
| Irrigation target $\theta_{target}$ | $\Phi_{sfc}$ | $\Phi_{sfc}$ | $\Phi_0$ | $\Phi_0$ |
| Water application location | Surface | Above the canopy | Surface | Surface with ponding |

\* $\Phi_{sfc}$ represents field capacity, \*\* $\Phi_0$ represents soil saturation.

R1C13: Figure 6 I would change the tick of y-axis because it is not clear to see the absolute number of irrigation water withdrawal.

Response: As per your suggestion, we have revised the tick of y-axis to better represent each model's absolute values of irrigation water withdrawal. Please see Figure 6 in the revised manuscript.

[Figure]

**Figure 6.** Comparison of irrigation water withdrawal simulated by CoLM and six global hydrological models participating in ISIMIP2a. (a) Annual total irrigation water withdrawal amounts in the United States as reported by the USGS, compared with simulations from CoLM (using both the irrig-unlim and irrig-lim schemes) and the six global hydrological models. (b) Comparison of reported and simulated irrigation water withdrawal for individual states, with Pearson correlation coefficient (*r*) and root mean square error (RMSE) for each simulation displayed.

---

## Author Comment (AC2)

**Response to Reviewers' Comments**

We greatly appreciate the editor and the reviewer for providing valuable and constructive comments on our manuscript (egusphere-2024-4093). We seriously considered each comment and revised the manuscript accordingly. The individual comments are addressed in the following response letter and the manuscript has been revised to accommodate the changes. Below are our detailed responses, with the comments from the editor and the reviewer in black followed by our responses in blue. Please note that the comments are marked with codes for brevity, such as R1C2 (Comment 2 from Reviewer 1).

R2C1: This study contributes to the field of irrigation modeling by incorporating an irrigation scheme into the Common Land Model (CoLM) by building upon established methodologies from existing literature. I believe this article has the potential for publication in Hydrology and Earth System Sciences (HESS). However, there is still much room for improvement in model validation. My comments are outlined below.

Response: We greatly appreciate the time and effort you have spent on our manuscript, as well as your encouraging feedback and constructive comments, which have been highly helpful in improving the overall quality of our work. We have provided a point-to-point response to each comment and made corresponding changes in the revised manuscript.

R2C2: Before introducing the "Two-way coupled irrigation water use module," it would be beneficial to provide a brief overview of the Common Land Model (CoLM)'s water and energy processes related to irrigation. This will help readers better understand CoLM's key mechanisms and how the new module integrates with them.

Response: As per your suggestion, we have added a detailed description of the water and energy processes in CoLM to provide better context for the integration of the two-way coupled irrigation module. Please see the Lines 173-199 in revised manuscript and Figure S1 in the revised supplement.

Lines 173-199: *"In CoLM2024, the 'patch' serves as the fundamental computational unit to account for land surface heterogeneity (Figure S1). Based on land type, patches are divided into five types: vegetation (including bare soil), urban, wetlands, glaciers, and water bodies. The vegetation patch is further classified into natural vegetation and crops, represented using the Plant Functional Type (PFT) approach. Under this framework, all natural vegetation within a grid cell is treated as a single*

*patch, sharing common soil thermal and moisture conditions while radiative and photosynthesis processes are simulated independently. When the crop model is activated, each crop type (distinguishing between rainfed and irrigated crops) is treated as an independent patch. This means that the calculations of soil moisture and thermal processes for each crop patch remain independent, without shared water and heat dynamics.*

*At each patch, the primary thermal processes include precipitation phase change, radiation transfer, temperature calculations for leaves, snow, and soil, turbulent exchange, etc. The key hydrological processes include canopy interception, evapotranspiration, surface runoff, infiltration, soil water vertical movement, subsurface runoff, groundwater, river routing, etc. Specifically, the two-big-leaf scheme is employed to compute radiation transfer, leaf temperature, photosynthesis and transpiration (Dai et al., 2004; Yuan et al., 2017). Surface turbulent exchange is simulated using similarity theory (Liu et al., 2022; Zeng and Dickinson, 1998). Soil and snow temperature are determined using the heat diffusion equation, considering only vertical exchange (Dai and Yuan, 2014). Canopy interception is calculated same as CoLM2014 with considering the leaf angle and precipitation phase (Dai and Yuan, 2014; Sellers et al., 1996). Soil water vertical movement is simulated by the Richards equation and Buckingham-Darcy's law with using the Campbell soil water characteristic curve scheme to close the Richards equation (Buckingham, 1907; Campbell, 1974; Richards, 1931). Surface and subsurface runoff are estimated using the SIMTOP approach (Niu et al., 2005). When the irrigation scheme is activated, irrigation water is applied to the canopy or top soil according to predefined irrigation methods and simulated irrigation amounts, thereby influencing the soil moisture and thermal processes within the irrigated patches."*

[Figure]

Figure S1. Diagram of the sub-grid structure in the Common Land Model.

R2C3: The irrigation system employed in this study is based on the soil moisture deficit method. The authors need to explain how this strategy is appropriate for the study region. Specifically, are the parameters $f_{irrig}$ and $f_{thresh}$ set to their default values? If so, using field capacity or saturation water amount as the target and threshold values for the root zone (1m) may be impractical, as it would theoretically demand an excessively large volume of water (as illustrated in Figure 3). While subsequent limitations might arise due to water availability constraints, this process remains inherently passive, as it is predicated on an initially overestimated assumption. If not, I suggest listing the value of the parameters.

Response: Yes, the parameters $f_{irrig}$ and $f_{thresh}$ were set to their default values in this study, as now stated in Line 241 of the revised manuscript.

We understand your concerns regarding the potential overestimation of irrigation demand due to parameter choices, including the target value, threshold value, and root zone depth. The soil moisture deficit method is a widely used approach for estimating irrigation requirements. We reviewed parameter settings in other hydrological and land surface models (Table R1) and found that most, like our study, adopt a uniform soil moisture threshold—either field capacity or saturation—while a few models (e.g., ORCHIDEE) account for crop-specific variations. As noted in our discussion, incorporating crop-specific thresholds is an important direction for future

improvement (Lines 825-829). Regarding root zone depth, CoLM does not simulate dynamic root growth, so we set a fixed depth of 1 m for all crops, consistent with the median crop root depth reported in previous studies (Table R1). While this assumption may not be realistic for shallow-rooted crops such as rice and soybean, large-scale root depth data remain scarce and are beyond the scope of this study. To avoid introducing additional uncertainties, we adopted the simplified approach commonly used in other models. We acknowledge this as a potential source of uncertainty and have revised the manuscript to discuss its implications (Lines 829-832).

Lines 825-832: *"However, the parameterization of certain key variables (e.g., target and threshold soil moisture levels) is overly simplified and does not account for variations among crop types. These parameters are adjustable, and their calibration could further enhance the model's accuracy in reproducing irrigation water use. Similarly, the fixed root depth of 1 m for all crops introduces additional uncertainty, potentially leading to overestimation or underestimation of irrigation demand. Incorporating dynamic root growth could better represent actual root zone depth based on crop-specific characteristics."*

**Table R1.** Review of scheme of irrigation in the literature.

| Model | Irrigation demand method* | Root depth | Target threshold |
|---|---|---|---|
| WaterGAP (Müller et al., 2014; 2021) | PET | -- | -- |
| PCR-GLOBWB (Sutanudjaja et al., 2018; Wada et al., 2014) | SMD | 0.5m-1.5m | no paddy: field capacity paddy: ponding 0.05m |
| H08 (Hanasaki et al., 2008;2018) | SMD | 0.15m | no paddy: 0.75* field capacity paddy: saturation |
| WBMplus (Grogan et al., 2022; Wisser et al., 2010) | SMD | 0.5m-1.5m | no paddy: field capacity paddy: ponding 0.05m |
| VIC (Haddeland et al., 2006; Zhou et al., 2016) | PET | -- | -- |
| MATSIRO (Pokhrel et al., 2012; 2015) | SMD | 1m | no paddy: 0.75* field capacity paddy: saturation |
| LPJmL (Jägermeyr et al., 2015) | SMD | 0.5m | saturation |
| ORCHIDEE (Arboleda-Obando et al., 2024) | SMD | 0.65m | saturation*crop coefficient |

| | | | |
|---|---|---|---|
| ELM
(Zhou et al., 2020) | SMD | 0.6m | between field capacity and saturation |
| CLM
(Leng et al., 2014; Yao et al., 2022) | SMD | 0.6m | between field capacity and saturation |
| Noah-MP
(He et al., 2023; Nie et al., 2018; Ozdogan et al., 2010) | SMD | 1m | field capacity or saturation |

[*] PET: potential evapotranspiration method; SMD: soil moisture deficit method.

R2C4: Model evaluation. Terrestrial Water Storage (TWS) anomaly is a crucial variable for model evaluation. Comparing this variable with GRACE satellite data would enhance the study's robustness and provide additional validation for the model's performance.

Response: Thank you for your valuable suggestion. We have incorporated an evaluation of the model's performance in simulating terrestrial water storage anomalies by comparing the results with GRACE satellite data. A detailed description of the GRACE dataset has been added (see Lines 439-442), and the validation results are presented in the revised manuscript (see Lines 656–683, Figure 9 and Figure S13).

Lines 439-442: *"For terrestrial water storage (TWS) validation, we utilized monthly terrestrial water storage anomaly data from the Gravity Recovery and Climate Experiment (GRACE) mission for the period 2002-2016, with a spatial resolution of 0.5 degree, provided by the NASA Jet Propulsion laboratory (Watkins et al., 2015; Wiese et al., 2016)."*

Lines 656-683:

*"**3.2.3 Evaluation of simulated terrestrial water storage anomalies**

To assess the model's ability to simulate the impact of irrigation on terrestrial water storage (TWS) dynamics, we compared the simulated monthly TWS anomalies with those derived from GRACE satellite products provided by the NASA Jet Propulsion laboratory. The results showed that incorporating the irrigation module, particularly under the irrig-lim scheme, improved the model's ability to capture both the interannual variability (Figure 9a) and seasonal patterns (Figure 9b) of TWS anomalies. Under the noirrig scheme, the simulated monthly TWS anomalies from 2002 to 2016 had a Pearson correlation of 0.25 with GRACE data and an RMSE of 6.75. In contrast, the irrig-lim scheme increased the correlation to 0.75 and reduced the RMSE to 5.13 (Figure 9a). The spatial distribution of Pearson correlation*

*coefficients between the simulations and GRACE data (Figure S13) further demonstrated a widespread improvement across the U.S., particularly in the Corn Belt.*

*The enhancement was even more pronounced in the simulation of seasonal TWS anomaly patterns. Without irrigation, the model underestimated seasonal variations, resulting in a pattern that deviated substantially from GRACE observations. This bias was effectively corrected in the irrig-lim scheme, where the Pearson correlation coefficient increased to 0.92 and the RMSE decreased to 3.44 (Figure 9b). However, none of the simulations captured the decline in GRACE-derived TWS anomalies during 2012 to 2016, likely due to groundwater depletion from irrigation (Rodell and Reager, 2023). This suggests that the model may require further validation and improvements in simulating irrigation-induced groundwater storage changes."*

[Figure]

**Figure 9.** Evaluation of simulated terrestrial water storage anomalies in irrigated region. (a) Time series of monthly terrestrial water storage anomalies from 2001 to 2016, simulated by CoLM (under the noirrig, irrig-unlim, and irrig-lim schemes) and derived from GRACE (JPL dataset). The Pearson correlation coefficient (r) and root mean square error (RMSE) between the simulations and GRACE data are indicated in the panel. (b) Climatological monthly terrestrial water storage anomalies averaged over 2001–2016, simulated by CoLM and derived from GRACE.

[Figure]

**Figure S13.** Comparison of observed and simulated monthly terrestrial water storage anomalies in the United States. (a) Spatial distribution of the Pearson correlation

coefficient (r) between GRACE-derived TWS anomalies (JPL dataset) and CoLM simulations under the noirrig scheme. (b–c) Same as (a) but for the irrig-unlim and irrig-lim schemes, respectively.

R2C5: Experimental Design. Consider creating a table that clearly delineates the differences between each experiment. Specifically, the table should highlight the scheme-specific variations in surface water and groundwater supply between the two irrigation experiments. This would help improve clarity and facilitate comparisons.

Response: Thanks for your suggestion. We have added a table to clearly delineate the differences between the experiments. Please see Table 1 in the revised manuscript.

**Table 1.** Experiment Configurations.

| Experiment | Management | Water limitation | Water sources |
|:---:|:---:|:---:|:---:|
| noirrig | Rainfed | NA | NA |
| irrig-unlim | Irrigated | No | NA |
| irrig-lim | Irrigated | Yes | Surface water and groundwater |

R2C6: Figure 7. Although the simulations of latent heat flux and sensible heat flux have improved, there is still a discrepancy compared to FLUXCOM data. Additionally, the fact that irrigation leads to an underestimation of temperature from June to September needs further discussion.

Response: Thank you for your insightful comments. We fully agree with your observations. While CoLM has improved the simulations of latent and sensible heat fluxes, it still exhibits notable uncertainties, particularly a systematic underestimation of evapotranspiration, even in non-cropland or non-irrigated areas (Figure S7). This issue reflects inherent limitations in CoLM itself, which are beyond the scope of this study. We have discussed the potential uncertainties introduced by CoLM's parameterization, including those related to evapotranspiration simulations, in the revised manuscript (see Lines 890–906).

We also acknowledge that incorporating irrigation led to a pronounced cooling effect, resulting in an underestimation of temperature from June to September, when irrigation volumes are highest. The impact of irrigation on land surface temperature is complex, as previous studies have shown that irrigation can induce both cooling and warming effects (Hu et al., 2019; McDermid et al., 2023; Thiery et al., 2017). The cooling effect primarily arises from increased evapotranspiration. However, irrigation can also enhance atmospheric water vapor content, leading to increased absorption of longwave radiation and potential impacts on cloud formation, which may contribute

to a warming effect (Dessler and Sherwood, 2009; Hu et al., 2019). Since this study employs offline land simulations, it does not account for irrigation-induced atmospheric feedbacks, which likely leads to an overestimation of the cooling effect and, consequently, an underestimation of temperature. This limitation is now explicitly discussed in the revised manuscript (Lines 753-762).

Lines 753–762: *"It is important to note that this study employs offline land simulations, which do not capture land-atmosphere interactions, potentially introducing biases in the estimated climate impacts. Previous studies have demonstrated that irrigation can induce both cooling and warming effects. While increased evapotranspiration contributes to cooling, irrigation can also enhance atmospheric water vapor content, leading to greater absorption of longwave radiation and potential cloud formation, resulting in warming (Dessler and Sherwood, 2009; Hu et al., 2019). These processes cannot be adequately represented in offline simulations, likely leading to an overestimation of irrigation-induced cooling and a subsequent underestimation of temperature (Figure 11g). Future studies should incorporate coupled land-atmosphere simulations to provide a more comprehensive assessment (Cook et al., 2015; Puma and Cook, 2010; Sacks et al., 2009)."*

R2C7: References. Some citations share the same author surname and publication year, which may cause linking issues. To resolve this, distinct labels (e.g.,2024a, 2024b) should be added, or additional author names can be included to differentiate them.

Response: Thanks for your suggestion. We have revised the references and citations accordingly by adding distinct labels. Please see Lines 96, 213, 348, and 509 in the revised manuscript.

R2C8: Figure S1. (1) Please confirm that the units in the figure (km3 -> km2 ?); (2) The color bar range needs to be adjusted. Due to at mid-latitudes (around 40°N), a 0.25°×0.25° grid cell covers approximately 770 km². Alternatively, you could use percentages to present the data for better clarity.

Response: Thank you for pointing out the unit error in the figure and for your constructive suggestion on improving the visualization. Following your suggestion, we have updated Figure S2 to present the data in percentages for better clarity.

[Figure]

(a) Crop Area

(b) Irrigated Area

**Figure S2.** Spatial distribution of crop and irrigated area percentages within the study region. (a) Percentage of crop area. (b) Percentage of irrigated area.

R2C9: Since the validation of simulated irrigation is obtained from the USGS. I suggest including some supporting text related to USGS and show total irrigation water withdrawals and categorized by surface and groundwater sources in Table format as a supplementary text.

Response: As per your suggestion, we have expanded the description of USGS irrigation water withdrawal data in the revised manuscript (Lines 479-481) and added a table in the supplementary materials (Table S3) to show total irrigation water withdrawals along with withdrawals from surface and groundwater sources.

**Table S3.** Observed and simulated irrigation water withdrawals ($km^3 \ yr^{-1}$).

| Sources | USGS | irrig-unlim | irrig-lim |
|---------|------|-------------|-----------|
| Total | 166.23 | 290.94 | 120.81 |
| Surface | 92.60 | NA | 37.78 |
| Groundwater | 73.63 | NA | 81.43 |

Reference:

Arboleda-Obando, P. F., Ducharne, A., Yin, Z., and Ciais, P.: Validation of a new global irrigation scheme in the land surface model ORCHIDEE v2.2, Geoscientific Model Development, 17, 2141–2164, https://doi.org/10.5194/gmd-17-2141-2024, 2024.

Dessler, A. E. and Sherwood, S. C.: A Matter of Humidity, Science, 323, 1020–1021, https://doi.org/10.1126/science.1171264, 2009.

Grogan, D. S., Zuidema, S., Prusevich, A., Wollheim, W. M., Glidden, S., and Lammers, R. B.: Water balance model (WBM) v.1.0.0: a scalable gridded global hydrologic model with water-tracking functionality, Geoscientific Model Development, 15, 7287–7323, https://doi.org/10.5194/gmd-15-7287-2022, 2022.

Haddeland, I., Skaugen, T., and Lettenmaier, D. P.: Anthropogenic impacts on continental surface water fluxes, Geophysical Research Letters, 33, https://doi.org/10.1029/2006GL026047, 2006.

Hanasaki, N., Kanae, S., Oki, T., Masuda, K., Motoya, K., Shirakawa, N., Shen, Y., and Tanaka, K.: An integrated model for the assessment of global water resources – Part 1: Model description and input meteorological forcing, Hydrology and Earth System Sciences, 12, 1007–1025, https://doi.org/10.5194/hess-12-1007-2008, 2008.

Hanasaki, N., Yoshikawa, S., Pokhrel, Y., and Kanae, S.: A global hydrological simulation to specify the sources of water used by humans, Hydrology and Earth System Sciences, 22, 789–817, https://doi.org/10.5194/hess-22-789-2018, 2018.

He, C., Valayamkunnath, P., Barlage, M., Chen, F., Gochis, D., Cabell, R., Schneider, T., Rasmussen, R., Niu, G.-Y., Yang, Z.-L., Niyogi, D., and Ek, M.: Modernizing the open-source community Noah with multi-parameterization options (Noah-MP) land surface model (version 5.0) with enhanced modularity, interoperability, and applicability, Geoscientific Model Development, 16, 5131–5151, https://doi.org/10.5194/gmd-16-5131-2023, 2023.

Hu, Z., Xu, Z., Ma, Z., Mahmood, R., and Yang, Z.: Potential surface hydrologic responses to increases in greenhouse gas concentrations and land use and land cover changes, International Journal of Climatology, 39, 814–827, https://doi.org/10.1002/joc.5844, 2019.

Jägermeyr, J., Gerten, D., Heinke, J., Schaphoff, S., Kummu, M., and Lucht, W.: Water savings potentials of irrigation systems: global simulation of processes and linkages, Hydrology and Earth System Sciences, 19, 3073–3091, https://doi.org/10.5194/hess-19-3073-2015, 2015.

Leng, G., Huang, M., Tang, Q., Gao, H., and Leung, L. R.: Modeling the Effects of Groundwater-Fed Irrigation on Terrestrial Hydrology over the Conterminous United States, Journal of Hydrometeorology, 15, 957–972, https://doi.org/10.1175/JHM-D-13-049.1, 2014.

McDermid, S., Nocco, M., Lawston-Parker, P., Keune, J., Pokhrel, Y., Jain, M., Jägermeyr, J., Brocca, L., Massari, C., Jones, A. D., Vahmani, P., Thiery, W., Yao, Y., Bell, A., Chen, L., Dorigo, W., Hanasaki, N., Jasechko, S., Lo, M.-H., Mahmood, R., Mishra, V., Mueller, N. D., Niyogi, D., Rabin, S. S., Sloat, L., Wada, Y., Zappa, L., Chen, F., Cook, B. I., Kim, H., Lombardozzi, D., Polcher, J., Ryu, D., Santanello, J., Satoh, Y., Seneviratne, S.,

Singh, D., and Yokohata, T.: Irrigation in the Earth system, Nature Reviews Earth & Environment, 1–19, https://doi.org/10.1038/s43017-023-00438-5, 2023.

Müller Schmied, H., Eisner, S., Franz, D., Wattenbach, M., Portmann, F. T., Flörke, M., and Döll, P.: Sensitivity of simulated global-scale freshwater fluxes and storages to input data, hydrological model structure, human water use and calibration, Hydrology and Earth System Sciences, 18, 3511–3538, https://doi.org/10.5194/hess-18-3511-2014, 2014.

Müller Schmied, H., Cáceres, D., Eisner, S., Flörke, M., Herbert, C., Niemann, C., Peiris, T. A., Popat, E., Portmann, F. T., Reinecke, R., Schumacher, M., Shadkam, S., Telteu, C.-E., Trautmann, T., and Döll, P.: The global water resources and use model WaterGAP v2.2d: model description and evaluation, Geoscientific Model Development, 14, 1037–1079, https://doi.org/10.5194/gmd-14-1037-2021, 2021.

Nie, W., Zaitchik, B. F., Rodell, M., Kumar, S. V., Anderson, M. C., and Hain, C.: Groundwater Withdrawals Under Drought: Reconciling GRACE and Land Surface Models in the United States High Plains Aquifer, Water Resources Research, 54, 5282–5299, https://doi.org/10.1029/2017WR022178, 2018.

Ozdogan, M., Rodell, M., Beaudoing, H. K., and Toll, D. L.: Simulating the Effects of Irrigation over the United States in a Land Surface Model Based on Satellite-Derived Agricultural Data, Journal of Hydrometeorology, 11, 171–184, https://doi.org/10.1175/2009JHM1116.1, 2010.

Pokhrel, Y., Hanasaki, N., Koirala, S., Cho, J., Yeh, P. J.-F., Kim, H., Kanae, S., and Oki, T.: Incorporating Anthropogenic Water Regulation Modules into a Land Surface Model, Journal of Hydrometeorology, 13, 255–269, https://doi.org/10.1175/JHM-D-11-013.1, 2012.

Pokhrel, Y. N., Koirala, S., Yeh, P. J.-F., Hanasaki, N., Longuevergne, L., Kanae, S., and Oki, T.: Incorporation of groundwater pumping in a global Land Surface Model with the representation of human impacts, Water Resources Research, 51, 78–96, https://doi.org/10.1002/2014WR015602, 2015.

Sutanudjaja, E. H., van Beek, R., Wanders, N., Wada, Y., Bosmans, J. H. C., Drost, N., van der Ent, R. J., de Graaf, I. E. M., Hoch, J. M., de Jong, K., Karssenberg, D., López López, P., Peßenteiner, S., Schmitz, O., Straatsma, M. W., Vannametee, E., Wisser, D., and Bierkens, M. F. P.: PCR-GLOBWB 2: a 5 arcmin global hydrological and water resources model, Geoscientific Model Development, 11, 2429–2453, https://doi.org/10.5194/gmd-11-2429-2018, 2018.

Thiery, W., Davin, E. L., Lawrence, D. M., Hirsch, A. L., Hauser, M., and Seneviratne, S. I.: Present-day irrigation mitigates heat extremes, Journal of Geophysical Research: Atmospheres, 122, 1403–1422, https://doi.org/10.1002/2016JD025740, 2017.

Wada, Y., Wisser, D., and Bierkens, M. F. P.: Global modeling of withdrawal, allocation and consumptive use of surface water and groundwater resources, Earth System Dynamics, 5, 15–40, https://doi.org/10.5194/esd-5-15-2014, 2014.

Wisser, D., Fekete, B. M., Vörösmarty, C. J., and Schumann, A. H.: Reconstructing 20th century global hydrography: a contribution to the Global Terrestrial Network- Hydrology (GTN-H), Hydrology and Earth System Sciences, 14, 1–24, https://doi.org/10.5194/hess-14-1-2010, 2010.

Yao, Y., Vanderkelen, I., Lombardozzi, D., Swenson, S., Lawrence, D., Jägermeyr, J., Grant, L., and Thiery, W.: Implementation and Evaluation of Irrigation Techniques in the Community Land Model, Journal of Advances in Modeling Earth Systems, 14, e2022MS003074, https://doi.org/10.1029/2022MS003074, 2022.

Zhou, T., Nijssen, B., Gao, H., and Lettenmaier, D. P.: The Contribution of Reservoirs to Global Land Surface Water Storage Variations, Journal of Hydrometeorology, 17, 309–325, https://doi.org/10.1175/JHM-D-15-0002.1, 2016.

Zhou, T., Leung, L. R., Leng, G., Voisin, N., Li, H.-Y., Craig, A. P., Tesfa, T., and Mao, Y.: Global Irrigation Characteristics and Effects Simulated by Fully Coupled Land Surface, River, and Water Management Models in E3SM, Journal of Advances in Modeling Earth Systems, 12, e2020MS002069, https://doi.org/10.1029/2020MS002069, 2020.